# Debiasing Graph Neural Networks via Learning Disentangled Causal Substructure

**Shaohua Fan[1,2]\*, Xiao Wang[1], Yanhu Mo[1], Chuan Shi[1][†], Jian Tang[2,3,4][†]**

[1] Beijing University of Posts and Telecommunications, China
[2] Mila - Québec AI Institute, Canada
[3] HEC Montréal, Canada
[4] CIFAR AI Research Chair

{fanshaohua, xiaowang, moyanhu, shichuan}@bupt.edu.cn, jian.tang@hec.ca

## Abstract

Most Graph Neural Networks (GNNs) predict the labels of unseen graphs by learning the correlation between the input graphs and labels. However, by presenting a graph classification investigation on the training graphs with severe bias, surprisingly, we discover that GNNs always tend to explore the spurious correlations to make decision, even if the causal correlation always exists. This implies that existing GNNs trained on such biased datasets will suffer from poor generalization capability. By analyzing this problem in a causal view, we find that disentangling and decorrelating the causal and bias latent variables from the biased graphs are both crucial for debiasing. Inspired by this, we propose a general disentangled GNN framework to learn the causal substructure and bias substructure, respectively. Particularly, we design a parameterized edge mask generator to explicitly split the input graph into causal and bias subgraphs. Then two GNN modules supervised by causal/bias-aware loss functions respectively are trained to encode causal and bias subgraphs into their corresponding representations. With the disentangled representations, we synthesize the counterfactual unbiased training samples to further decorrelate causal and bias variables. Moreover, to better benchmark the severe bias problem, we construct three new graph datasets, which have controllable bias degrees and are easier to visualize and explain. Experimental results well demonstrate that our approach achieves superior generalization performance over existing baselines. Furthermore, owing to the learned edge mask, the proposed model has appealing interpretability and transferability.[3]

## 1 Introduction

Graph Neural Networks (GNNs) have exhibited powerful performance on graph data with various applications [17, 35, 13, 9, 8]. One major category of applications are the graph classification task, such as molecular graph property prediction [15, 20, 44], superpixel graph classification [14], and social network category classification [46, 44]. It is well known that graph classification is usually determined by a relevant substructure, but not the whole graph structure [43, 26, 45]. For example, for MNIST superpixel graph classification task, the digit subgraphs are causal (*i.e.*, deterministic) for labels [36]. The mutagenic property of a molecular graph depends on the functional groups (*i.e.*,

---

\*This work was done when the first author was a visiting student at Mila.

[†]Corresponding authors.

[3]Code and data are available at: https://github.com/googlebaba/DisC.

36th Conference on Neural Information Processing Systems (NeurIPS 2022).

nitrogen dioxide ($NO_2$)), rather than the irrelevant patterns (*i.e.*, carbon rings) [27]. Therefore, it is a fundamental requirement for GNNs to identify causal substructures, so as to make correct prediction.

Ideally, when the graphs are unbiased, *i.e.*, only the causal substructures are related with the graph labels, the GNNs are able to utilize such substructure to predict the labels. However, due to the uncontrollable data collection process, the graphs are inevitably biased, *i.e.*, existing meaningless substructures spuriously correlates with labels. Taking a colored MNIST superpixel graph dataset in Sec. 3.1 as an example (illustrated in Fig. 1(a)), each category of digit subgraphs mainly correspond to one kind of color background subgraphs, *e.g.*, digit 0 subgraph is related with red background subgraph. Therefore, the color background subgraph will be treated as bias information, which highly correlates with labels but does not determines them in the training set. Under this situation, *will GNNs still stably utilize the causal substructure to make decision?*

To investigate the impact of bias on GNNs, we conduct an experimental investigation to demonstrate the impact of bias (especially in the severe bias scenarios) on the generalization capability of GNNs (Sec. 3.1). We find that GNNs actually utilize both bias and causal substructures to make prediction. However, with severer bias correlation, even bias substructure still could not exactly determine labels like causal substructure, GNNs majorly utilize bias substructure as shortcuts to make prediction, causing a large generalization performance degradation. Why this happens? We analyze the data-generating process and model prediction mechanism behind the graph classification using a causal graph (Sec. 3.2). The casual graph illustrates that the observed graphs are generated by the causal and bias latent variables and existing GNNs could not distinguish the causal substructure from entangled graphs. *How can we disentangle the causal and bias substructures from observed graphs, so that GNNs can only utilize the causal substructures to make stable prediction when severe bias appears?*

To address the question, two challenges need to be faced. 1) How to identify the causal substructure and bias substructure in the severe biased graphs? In the severe bias scenarios, bias substructure will be "easier to learn" for GNNs and finally dominate the prediction. Using the normal cross-entropy loss, like DIR [39], could not fully capture such aggressive property of bias. 2) How to extract the causal substructure from an entangled graph? The statistically causal substructure is usually determined by the global property of the entire graph population, rather than a single graph. When extracting causal substructure from a graph, we need to establish the relations among all the graphs.

In this paper, we propose a novel debiasing framework for GNNs via learning **Dis**entangled **C**ausal substructure, called **DisC**. Given an input biased graph, we propose to explicitly filter edges into causal and bias subgraphs by a parameterized edge mask generator, whose parameters are shared across entire graph population. As a result, the edge masker is naturally capable to specify the importance for each edge and extract causal and bias subgraphs from a global view of the entire observations. Then, a "casual"-aware (weighted cross-entropy) loss and a "bias"-aware (generalized cross-entropy) loss are respectively utilized to supervise two functional GNN modules. Based on the supervision, the edge mask generator could generate corresponding subgraphs and the GNNs could encode corresponding subgraphs into their disentangled embeddings. With the disentangled embeddings, we randomly permute the latent vectors extracted from different graphs to generate more unbiased counterfactual samples in embedding space. The new generated samples still contain both causal and bias information, while their correlation has been decorrelated. In this time, there is only correlation between causal variables with labels, so that the model could concentrate on the true correlation between the causal subgraphs and labels. Our major contributions are as follows:

- To our knowledge, we first study the generalization problem of GNNs in a more challenging yet practical scenario, i.e., the graphs are with severe bias. We systematically analyze the bias impact on GNNs from both experimental study and causal analysis. We find that the bias substructure, compared with causal substructure, is much easier to dominate the training of GNNs.

- To debias GNNs, we develop a novel GNN framework for disentangling causal substructure, which is flexible to build upon various GNNs for improving generalization ability while enjoying inherent interpretability, robustness and transferability.

- We construct three new datasets with various properties and controllable bias degrees, which can better benchmark the new problem. Our model outperforms the corresponding base models with a large margin (from 4.47% to 169.17% average improvements). Various investigation studies demonstrate that our model could discover and leverage causal substructure for prediction.

## 2 Related Works

**Generalization of GNNs in wild environments.** Most existing GNN methods are proposed under the IID hypothesis, *i.e.*, training and testing set are independently sampled from the identical distribution [34, 17, 35, 13, 24]. However, in reality, thus ideal assumption is hard to be satisfied. Recently, several methods have been proposed to improve the generalization ability of GNNs in wild OOD environments. Several works [29, 7, 38] study the OOD problem of node classification. For OOD graph classification task, StableGNN [6] propose to learn the stable causal relationship in graphs. OOD-GNN [22] propose to constrain each dimension of learned embedding to be independent. DIR [39] discovers the invariant rationales for generalizing GNNs. Although they have achieved better OOD performance, they are not designed for the datasets with severe bias, which is more challenging for guaranteeing the generalization ability of GNNs.

**Disentangled graph neural networks.** Recently, there are a couple of methods that study the disentangled GNNs. DisenGCN [28] utilizes neighbourhood routing mechanism to divide the neighbours of the node into several mutually exclusive parts. IPGDN [25] promotes DisenGCN by constraining the different parts of the embedding feature to be independent. DisenGCN and IPGDN are node-level disentanglement, thus FactorGCN [42] considers the whole graph information and disentangles the target graph into several factorized graphs. Despite results of the previous works, they do not consider disentangling the causal and bias information for graphs.

**General debiasing methods.** Recently, debiasing problem has drawn much attention in machine learning community [16, 23, 33, 1, 2, 11]. One category of these methods is pre-defining a certain bias type explicitly to mitigate [16, 23, 33, 1, 37]. For example, Wang et al. [37] and Bahng et al. [1] design a texture- and color-guided model to adversarially train a debiased neural network against the biased one. Instead of defining certain types of bias, recent approaches [30, 5, 21] rely on the straightforward assumption that models are prone to exploit the bias as shortcuts to make prediction [10]. In the line with the recent studies, our study belongs to the second category. However, most of existing methods are designed for image datasets and could not effectively extract causal substructure from graph data. Distinctly, we first study the severe bias problem on graph data, and our method could effectively extract causal substructure from graph data.

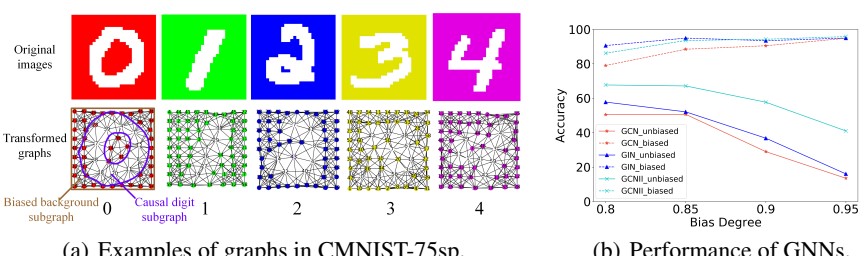

(a) Examples of graphs in CMNIST-75sp.   (b) Performance of GNNs.

Figure 1: Example graphs of CMNIST-75sp and the performance of GNNs on this dataset.

## 3 Preliminary Study and Analysis

In this section, we first illustrate the existing GNNs tend to exploit the bias substructure as shortcuts for prediction through a motivating experiment. Then we analyze the prediction process of GNNs in causal view. Based on this causal view, it motivates our solution to relieve the impact of bias.

### 3.1 Motivating Example

To measure the generalization ability of GNNs with the effect of bias, we construct a graph classification dataset with controllable bias degrees, called CMNIST-75sp. We first construct a biased MNIST image dataset like [1], where each category of digit highly correlates with a pre-defined color in their background. For example, in the training set, 90% of 0 digits are with red background (*i.e.*, biased samples), and remaining 10% images are with random background color (*i.e.*, unbiased samples), whose the bias degree is 0.9 in this situation. We consider four bias degrees $\{0.8, 0.85, 0.9, 0.95\}$.

For the testing set, we construct both biased testing set and unbiased testing set. The biased testing set has the same bias degree with training set, aiming to measure the extent of models relying on bias. The unbiased testing set, where the digit labels uncorrelate with the background colors, aims to test whether the model could utilize the inherent digit signals for prediction. Note that training set and testing set have the same pre-defined color set. Then, we convert the biased MNIST images into superpixel graphs with at most 75 nodes each graph using [18], where the edges are constructed by the KNN method based on the 2D coordinates of superpixels and node features are the concatenation of coordinates and average color of superpixels. Each graph is labeled by its digit class, so that its digital subgraph is deterministic for label and background subgraph is spuriously correlated with labels but not deterministic. The examples of graphs are illustrated in Fig. 1(a).

We perform three popular GNN methods: GCN [17], GIN [41], and GCNII [3] on CMNIST-75sp and the results are shown in Fig. 1(b). The same color of dashed line and solid line represent the results of the corresponding methods on the biased testing set and the unbiased testing set respectively. Overall, the GNNs achieve much better performance on biased testing set than unbiased testing set. The phenomenon indicates that although GNNs could still learn some causal signals for prediction, the unexpected bias information is also being utilized for prediction. More specifically, with bias degree becoming larger, the performance of GNNs on biased testing set is increased and the value of accuracy is nearly in line with the bias degree, while the performance on unbiased testing drops dramatically. Hence, although causal substructure could determine labels perfectly, in severe bias scenarios, the GNNs lean to utilize the easier to learn bias information to make prediction rather than the inherent causal signals, and bias substructure will finally dominate the prediction.

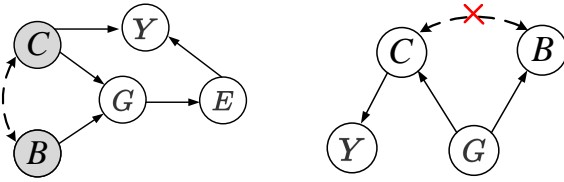

(a) SCM of the union of the data generation and the GNNs' prediction process.

(b) SCM of our debiasing GNN existing method.

Figure 2: SCMs. Grey and white variables represent unobserved and observed variables, respectively.

## 3.2 Problem Analysis

Debiasing GNNs for unbiased prediction requires understanding the natural mechanisms of graph classification task. We present a causal view of the union of the data-generating process and the model prediction process behind the task. Here we formalize the causal view as a Structure Causal Model (SCM) or causal graph [12, 31] by inspecting on the causalities among five variables: unobserved causal variable $C$, unobserved bias variable $B$, observed graph $G$, graph embedding $E$, and ground truth label / prediction $Y$[4]. Fig. 2(a) illustrates the SCM, where each link denotes a causal relationship.

- $C \rightarrow G \leftarrow B$. The observed graph data is generated by two unobserved latent variables: the causal variable $C$ and the bias variable $B$, such as digit subgraphs and background subgraphs in the CMNIST-75sp dataset. And all bellow relations are illustrated by CMNIST-75sp.

- $C \rightarrow Y$. This link means that the causal variable $C$ is the only endogenous parent to determine the generation of ground-truth label $Y$. For example, $C$ is the oracle digit subgraph, which exactly explains why the label is labeled as $Y$.

- $C \leftarrow\!\!\rightarrow B$. This link indicates the spurious correlation between $C$ and $B$. Such probabilistic dependencies is usually caused by the direct cause or unobserved confounder [32]. Here we do not distinguish these scenarios and only observe the spurious correlation between $B$ and $C$, such as the spurious correlation between the color background subgraphs and digit subgraphs.

---

[4]We use variable Y for both the ground-truth labels and prediction, as they are optimized to be the same.

- $G \rightarrow E \rightarrow Y$. Existing GNNs usually learn the graph embedding $E$ based on the observed graph $G$ and make the prediction $Y$ based on the learned embedding $E$.

According to the SCM, GNNs will utilize both information to make prediction. As bias substructure (*e.g.*, background subgraph) usually has simpler structure than meaningful causal substructure (*e.g.*, digit subgraph), if GNN utilizes such simple substructure, it could achieve low loss very fast. Hence, GNN inclines to utilizes bias information when most graphs are biased. Based on the SCM in Fig. 2(a), according to $d$-connection theory [31] (see App. A): two variables are dependent if they are connected by at least one unblocked path, we could find two paths that would induce the spurious correlation between the bias variable $B$ and label $Y$: **(1) B $\rightarrow$ G $\rightarrow$ E $\rightarrow$ Y** and **(2) B $\leftrightarrow$ C $\rightarrow$ Y**. To make the prediction $Y$ being uncorrelated with the bias $B$, we need to intercept the two unblocked paths. For this purpose, we propose to debias GNNs in causal view, as in Fig. 2(b).

- $C \leftarrow G \rightarrow B$ and $C \rightarrow Y$. To intercept the path (1), we should disentangle the latent variables $C$ and $B$ from the observed graph $G$ and make prediction only based on the causal variable $C$.

- $C \leftarrow \!\!\times\!\!\rightarrow B$. To intercept the path (2), as we cannot change the link between $C$ and $Y$, one possible solution is to make $C$ and $B$ uncorrelated.

# 4    Methodology

Motivated by the above causal analysis, in this section, we present our proposed debiasing GNN framework **DisC**, to remove the spurious correlation. The overall framework is shown in Fig. 3. First, an edge mask generator is learnt to mask the edges of original input graphs into causal subgraphs and bias subgraphs. Second, two separate GNN modules with their corresponding masked subgraphs are trained to encode corresponding causal substructure and bias substructure into disentangled representations, respectively. Last, after the disentangled representations are well-trained, we permute the bias representations among the training graphs to generate counterfactual unbiased samples, so that the correlation between causal representations and bias representations is removed.

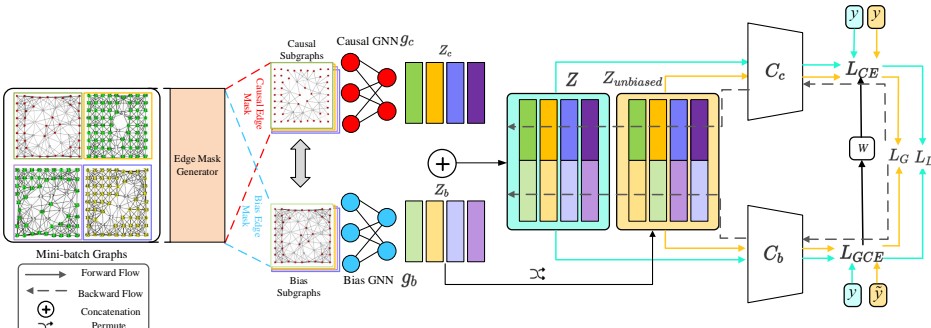

Figure 3: The overall framework of DisC.

## 4.1    Causal and Bias Substructure Generator

Given a mini-batch of biased graphs $\mathcal{G} = \{G_1, \cdots, G_n\}$, our idea is that: we take a collection of graph instances and design a generative probabilistic model to learn to mask the edges into causal subgraph or bias subgraph. Particularly, given a graph $G = \{\mathbf{A}, \mathbf{X}\}$, where $\mathbf{A}$ is the adjacency matrix and $\mathbf{X}$ is the node feature matrix, we utilize a multi-layer perceptron (MLP) upon the concatenation of node features $\mathbf{x}_i$ of node $i$ and $\mathbf{x}_j$ of node $j$ to measure the importance of edge $(i, j)$ for causal subgraph:

$$\alpha_{ij} = \text{MLP}([\mathbf{x}_i, \mathbf{x}_j]). \tag{1}$$

Then a sigmoid function $\sigma(\cdot)$ is employed to project $\alpha_{ij}$ into the range of $(0,1)$, which indicates the probability of edge $(i, j)$ being the edge in the causal subgraph as follows:

$$c_{ij} = \sigma(\alpha_{ij}). \tag{2}$$

Naturally, we could get the probability of edge $(i, j)$ being the edge in the bias subgraph by: $b_{ij} = 1 - c_{ij}$. Now we could construct the causal edge mask $\mathbf{M}_c = [c_{ij}]$ and bias edge mask $\mathbf{M}_b = [b_{ij}]$. Finally, we decompose the original graph $G$ into causal subgraph $G_c = \{\mathbf{M}_c \odot \mathbf{A}, \mathbf{X}\}$ and bias subgraph $G_b = \{\mathbf{M}_b \odot \mathbf{A}, \mathbf{X}\}$. Intuitively, the edge mask could highlight different part of structure information of original graphs, thus GNNs built on the different subgraphs could encode different parts of graph information. Moreover, the mask generator has two advantages. (1) **Global view:** In individual graph level, the mask generator (*i.e.*, MLP), whose parameters are shared by all the edges in a graph, take a global view of all the edges in a graph, which enables us to identify community in graph. It is well known that the effect of an edge cannot be judged independently, because edges usually collaborate with each other, forming a community, to make prediction. Thus, it is critical to evaluate an edge in a global view. In whole graph population level, the mask generator takes a global view of all the graphs in the training set, which enables us to identify causal/bias subgraph. Particularly, as the causal/bias is the statistical information in the population level, it is necessary to view all the graphs to identify the causal/bias substructure. Considering both such coalition effects and population-level statistical information, the generator is able to measure the importance of edges more accurately. (2) **Generalization**: The mask generator can generalize the mechanism of mask generation to new graphs without retraining, so it is capable and efficient to prune unseen graphs.

## 4.2 Learning Disentangled Graph Representations

Given $G_c$ and $G_b$, how to ensure they are causal subgraph and bias subgraph, respectively? Inspired by [21], our approach simultaneously trains a pair of GNNs $(g_b, g_c)$ with linear classifiers $(C_b, C_c)$ as follows: (1) Motivated by the observation in Sec. 3.1 that bias substructure is easier to learn, we utilize a bias-aware loss to train a bias GNN $g_b$ and a bias classifier $C_b$ and (2) in contrast, we train a causal GNN $g_c$ and a causal classifier $C_c$ on the training graphs that the bias GNN struggles to learn. Next, we would present each component in detail.

As shown in Fig. 3, GNN $g_c$ and $g_b$ embed the corresponding subgraphs into causal embedding $z_c = g_c(G_c; \gamma_c)$ and bias embedding $z_b = g_b(G_b; \gamma_b)$, respectively, where $\gamma$ is the parameters of GNNs. Subsequently, concatenated vector $z = [z_c; z_b]$ is fed into linear classifiers $C_c$ and $C_b$ to predict the target label $y$. To train $g_b$ and $C_b$ as bias extractor, we utilize the generalized cross entropy (GCE) [47] loss to amplify the bias of the bias GNN and classifier:

$$GCE(C_b(z; \alpha_b), y) = \frac{1 - C_b^y(z; \alpha_b)^q}{q}, \tag{3}$$

where $C_b(z; \alpha_b)$ and $C_b^y(z; \alpha_b)$ are softmax output of the bias classifier and its probability belonging to the target category $y$, respectively, and $\alpha$ is the parameters of classifier. Here $q \in (0, 1]$ is a hyperparameter that controls the degree of amplifying bias. Given $\theta_b = [\gamma_b, \alpha_b]$, the gradient of the GCE loss up-weights the gradient of the standard cross entropy (CE) loss for the samples with a high confidence $C_b^y$ of predicting the correct target category as follows:

$$\frac{\partial GCE(C_b(z; \alpha_b), y)}{\partial \theta_b} = (C_b^y)^q \frac{\partial CE(C_b(z; \alpha_b), y)}{\partial \theta_b}. \tag{4}$$

Therefore, compared with CE loss, GCE loss will amplify the gradients of $\theta_b$ on samples by the confidence score $(C_b^y)^q$. Based on our observation that the bias information is usually easier to be learned, so the biased graphs will have higher $(C_b^y)^q$ than unbiased graphs. Therefore, the model $g_b$ and $C_b$ trained by GCE loss will focus on bias information and finally get the bias subgraph. Note that, to ensure that $C_b$ predicts target labels mainly based on this $z_b$, the loss from $C_b$ is not backpropagated to $g_c$, *i.e.*, only update $\theta_b$ in Eq. (4), and vice versa.

Meanwhile, we also train a causal GNN simultaneously with the weighted CE loss. The graphs with high CE loss from $C_b$ can be regarded as the unbiased samples compared with the samples with low CE loss. In this regard, we could obtain the unbias score of each graph as

$$W(z) = \frac{CE(C_b(z), y)}{CE(C_c(z), y) + CE(C_b(z), y)}. \tag{5}$$

Large value of $W$ implies the graph is an unbiased sample, hence we could use these weights to reweight the loss of these graphs to train $g_c$ and $C_c$, enforcing them to learn the unbiased information. Thus, the objective function for learning disentangled representation is:

$$L_D = W(z)CE(C_c(z), y) + GCE(C_b(z), y). \tag{6}$$

### 4.3 Counterfactual Unbiased Sample Generation

Until now, we have achieved the first goal analyzed in Sec. 3.2 that is the disentanglement of causal and bias substructures. Next, we will show how to achieve the second goal that makes the causal variable $z_c$ and bias variable $z_b$ uncorrelated. Although we have disentangled causal and bias information, they are disentangled from the biased observed graphs. Hence, there will exist statistical correlation between causal and bias variables inheriting from the biased observed graphs. To further decorrelate $z_c$ and $z_b$, according to the causal relation of data-generating process: $C \rightarrow G \leftarrow B$, we propose to generate the counterfactual unbiased samples in embedding space by swapping $z_b$. More specifically, we randomly permute bias vectors in each mini-batch and obtain $z_{unbiased} = [z_c; \hat{z}_b]$, where $\hat{z}_b$ represents the randomly permuted bias vectors of $z_b$. As $z_c$ and $\hat{z}_b$ in $z_{unbiased}$ are randomly combined from different graphs, they will have much less correlation than $z = [z_c; z_b]$ where both are from the same graph. To make $g_b$ and $C_b$ still focus on the bias information, we also swap label $y$ as $\hat{y}$ along with $\hat{z}_b$, so that the spurious correlation between $\hat{z}_b$ and $\hat{y}$ still exists. With the generated unbiased samples, we utilize the following loss function to train two GNN modules:

$$L_G = W(z)CE(C_c(z_{unbiased}), y) + GCE(C_b(z_{unbiased}), \hat{y}), \tag{7}$$

Together with the disentanglement loss, total loss function is defined as:

$$L = L_D + \lambda_G L_G, \tag{8}$$

where $\lambda_G$ is a hyperparameter for weighting the importance of generation component. Moreover, training with more diverse samples would also benefit with better generalization on unseen testing scenarios. Our approach is summarized in App. B. Note that, as we need well-disentangled representations to generate the high-quality unbiased samples, in the early stage of training, we only train the model with $L_D$. After certain epochs, we train the model with $L$.

## 5 Experiment

**Datasets.** We construct three datasets with various properties and bias ratios to benchmark this new problem, where the datasets have clear causal subgraphs making the results explainable. Following CMNIST-75sp introduced in Sec. 3.1, we use the similar way to construct CFashion-75sp and CKuzushiji-75sp datasets based on the Fashion-MNIST [40] and Kuzushiji-MNIST [4] datasets. As the causal subgraphs of these two datasets are more complicated (fashion product and hiragana characters), they are more challenging. Due to the page limits, here we set bias degrees as $\{0.8, 0.9, 0.95\}$. We report the main results on unbiased test sets. Details are in App. C.1.

**Baselines and experimental setup.** As DisC is a general framework which could be built on various base GNN models, we select three popular GNNs: GCN [17], GIN [41], and GCNII [3]. The corresponding models are termed as $\text{DisC}_{GCN}$, $\text{DisC}_{GIN}$ and $\text{DisC}_{GCNII}$, respectively. Hence, base models are the most straight baselines. Another kind of baselines are the causal-inspired GNN method DIR [39] and StableGNN [6]. We also compare against a general debiasing method LDD [21] by replacing its encoder with GNNs. Graph Pooling method DiffPool [44] and graph disentangling method FactorGCN [42] are also compared. To keep fair comparison, our model uses the same GNN architecture and hyperparameters with the corresponding base model. All the experiments are run 4 times with different random seeds and we report the accuracy and the standard error. More details are in App. C.2.

### 5.1 Quantitative Evaluation

**Main results.** The overall results are summarized in Table 1, and we have following observations:

(1) DisC has much better generalization ability than base models. DisC outperforms the corresponding base model consistently with a large margin. With heavier biases, our model achieves larger improvements over base models. Specifically, for CMNIST-75sp, CFashion-75sp and CKuzushiji-75sp with smaller bias degree (*i.e.*, 0.8), our model achieves 40.02%, 4.47% and 29.82% average improvements over corresponding base models, respectively. Surprisingly, with severer biases (0.9 and 0.95), DisC achieves 169.17%, 14.67% and 49.35% average improvements over base models on three datasets, respectively. It indicates that the proposed method is a general framework helping existing GNNs against the negative impact of bias.

(2) DisC significantly outperforms existing debiasing methods. We notice that DIR could not achieve satisfying results. The reason is that DIR utilizes CE loss to extract bias information, which could not fully capture the property of bias in severe bias scenarios. And DIR sets one fixed threshold to spilt subgraphs, which is suboptimal. StableGNN outperforms their base model DiffPool and achieve competitive results, indicating the effectiveness of their proposed causal variable distinguishing regularizer. However, their framework adjusts data distribution based on the original dataset, it is hard to generate unbiased distribution when the unbiased samples are scarce. DisC could generate more unbiased samples based on the disentangled representations. Moreover, LDD is a general debiasing method which is not designed for graph data. DisC outperforms corresponding LDD variants with average 23.15%, indicating that the seamless joint of global-population-aware edge masker with debiasing disentangle framework is very effective.

Table 1: Graph classification accuracy evaluated on unbiased testing sets, which have same color (bias) set with training set. The best performance within each base model variant is in bold.

| Dataset | CMNIST-75sp | | | CFashion-75sp | | | CKuzushiji-75sp | | |
|---|---|---|---|---|---|---|---|---|---|
| Bias | 0.8 | 0.9 | 0.95 | 0.8 | 0.9 | 0.95 | 0.8 | 0.9 | 0.95 |
| FactorGCN [42] | $72.30_{\pm1.18}$ | $62.35_{\pm5.07}$ | $42.50_{\pm4.91}$ | $61.23_{\pm1.11}$ | $53.50_{\pm1.29}$ | $45.78_{\pm2.40}$ | $42.87_{\pm1.19}$ | $32.35_{\pm2.79}$ | $23.87_{\pm0.12}$ |
| DiffPool [44] | $73.79_{\pm0.02}$ | $66.45_{\pm0.78}$ | $47.12_{\pm1.04}$ | $62.82_{\pm0.53}$ | $57.50_{\pm0.39}$ | $50.86_{\pm0.20}$ | $45.46_{\pm0.65}$ | $36.18_{\pm0.19}$ | $27.45_{\pm0.26}$ |
| DIR [39] | $9.98_{\pm0.33}$ | $9.96_{\pm0.23}$ | $10.03_{\pm0.27}$ | $13.02_{\pm1.92}$ | $12.80_{\pm1.67}$ | $11.98_{\pm1.41}$ | $10.35_{\pm0.32}$ | $10.72_{\pm0.27}$ | $10.59_{\pm0.46}$ |
| StableGNN [6] | $77.65_{\pm1.64}$ | $68.87_{\pm1.74}$ | $51.33_{\pm0.87}$ | $64.03_{\pm0.29}$ | $58.26_{\pm0.09}$ | $51.46_{\pm0.39}$ | $49.41_{\pm0.09}$ | $39.30_{\pm0.12}$ | $28.26_{\pm0.14}$ |
| LDD$_{GCN}$ [21] | $64.95_{\pm1.22}$ | $56.65_{\pm2.18}$ | $46.83_{\pm2.88}$ | $63.85_{\pm1.17}$ | $64.30_{\pm0.89}$ | $62.28_{\pm0.48}$ | $42.38_{\pm0.33}$ | $38.75_{\pm0.49}$ | $33.08_{\pm0.59}$ |
| LDD$_{GIN}$ [21] | $64.88_{\pm1.45}$ | $50.59_{\pm1.07}$ | $31.23_{\pm2.48}$ | $64.65_{\pm0.63}$ | $57.10_{\pm0.43}$ | $53.38_{\pm0.47}$ | $37.83_{\pm0.54}$ | $28.97_{\pm0.18}$ | $22.13_{\pm0.34}$ |
| LDD$_{GCNII}$ [21] | $78.03_{\pm0.66}$ | $69.53_{\pm0.96}$ | $51.05_{\pm3.87}$ | $50.63_{\pm1.79}$ | $54.09_{\pm2.54}$ | $57.93_{\pm0.88}$ | $48.70_{\pm1.98}$ | $41.59_{\pm1.07}$ | $33.93_{\pm0.71}$ |
| GCN [17] | $50.43_{\pm4.13}$ | $28.97_{\pm4.4}$ | $13.50_{\pm1.38}$ | $63.60_{\pm0.53}$ | $57.22_{\pm0.93}$ | $47.69_{\pm0.42}$ | $38.45_{\pm1.1}$ | $28.35_{\pm0.79}$ | $20.70_{\pm0.88}$ |
| DisC$_{GCN}$ | $\mathbf{82.60_{\pm0.93}}$ | $\mathbf{78.14_{\pm2.14}}$ | $\mathbf{63.47_{\pm5.65}}$ | $\mathbf{66.85_{\pm1.11}}$ | $\mathbf{65.33_{\pm4.70}}$ | $\mathbf{63.93_{\pm1.50}}$ | $\mathbf{55.53_{\pm2.29}}$ | $\mathbf{48.13_{\pm2.59}}$ | $\mathbf{36.63_{\pm1.73}}$ |
| GIN [41] | $57.75_{\pm0.78}$ | $36.78_{\pm5.55}$ | $16.04_{\pm1.14}$ | $64.25_{\pm0.46}$ | $58.03_{\pm0.40}$ | $49.74_{\pm0.60}$ | $41.83_{\pm0.78}$ | $30.09_{\pm0.87}$ | $21.18_{\pm1.63}$ |
| DisC$_{GIN}$ | $\mathbf{82.10_{\pm1.50}}$ | $\mathbf{74.90_{\pm1.81}}$ | $\mathbf{58.58_{\pm4.24}}$ | $\mathbf{67.10_{\pm1.07}}$ | $\mathbf{59.90_{\pm1.31}}$ | $\mathbf{55.80_{\pm0.36}}$ | $\mathbf{55.18_{\pm1.00}}$ | $\mathbf{41.75_{\pm0.81}}$ | $\mathbf{30.25_{\pm1.63}}$ |
| GCNII [3] | $69.70_{\pm1.73}$ | $57.68_{\pm1.68}$ | $41.00_{\pm3.75}$ | $\mathbf{66.68_{\pm0.59}}$ | $60.58_{\pm0.28}$ | $53.18_{\pm0.08}$ | $48.53_{\pm0.25}$ | $36.23_{\pm0.20}$ | $25.60_{\pm0.76}$ |
| DisC$_{GCNII}$ | $\mathbf{79.50_{\pm2.48}}$ | $\mathbf{76.00_{\pm1.90}}$ | $\mathbf{60.54_{\pm5.33}}$ | $66.47_{\pm1.77}$ | $\mathbf{65.48_{\pm0.70}}$ | $\mathbf{61.75_{\pm0.27}}$ | $\mathbf{54.90_{\pm1.30}}$ | $\mathbf{44.73_{\pm1.55}}$ | $\mathbf{36.95_{\pm0.70}}$ |

**Ablation studies.** To validate the importance of each module in our method, in Fig. 4, we conduct ablation studies on our variants (w.o. G means without the sample generation module) and the related variants of LDD. The major difference between DisC/w.o. G with LDD /w.o. G is the edge mask module. In most cases, DisC/w.o. G significantly outperforms LDD /w.o. G, indicating the necessity of learning edge mask for graph data. And DisC which has counterfactual sample generation module could further boost the performances based on the disentangled embeddings of DisC/w.o. G. However, LDD seldomly outperforms LDD /w.o. G or even achieves worse performances. That is, generating high-quality counterfactual samples needs well-disentangled causal and bias embeddings. If embeddings are not well-disentangled, counterfactual samples may act as noisy samples, which would prevent models from achieving further improvement. The edge masker could help the model generate well-disentangled embeddings, which is crucial for overall performance.

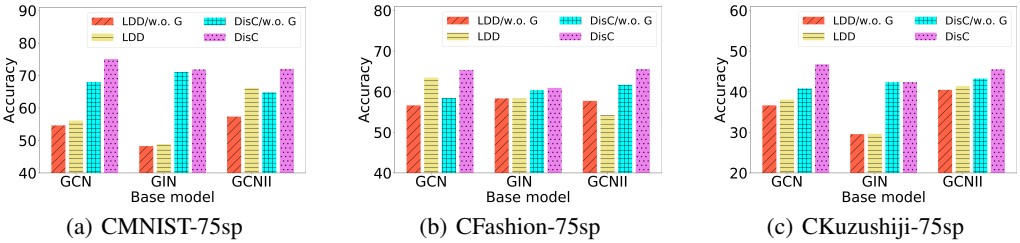

(a) CMNIST-75sp      (b) CFashion-75sp      (c) CKuzushiji-75sp

Figure 4: Ablation studies of the DisC vs. LDD average over three bias degrees of each dataset.

**Robustness on unseen bias.** Table 2 reports the results of DisC compared with its corresponding base models on testing set with unseen bias, $i.e.$, the pre-defined color (bias) sets of training set and testing set are disjoint. The performances of base models further drop compared with the results on seen bias scenario in Table 1. However, our model still achieves very stable performances, fully demonstrating the generalization ability of our model on agnostic bias scenario.

**Hyperparameter experiments** Fig. 5 is the hyperparameter experiments of the degree of amplifying bias $q$ in GCE loss and the importance of generation component $\lambda_G$. For $q$, we fix $\lambda_G = 10$ and vary $q$ from $\{0.1, 0.3, 0.5, 0.7, 0.9\}$. For $\lambda_G$, we fix $q = 0.7$ and vary $\lambda_G$ from $\{1, 5, 10, 15\}$. From the results, we can see that our model achieves stable performance across different values of $q$ and

Table 2: The results on unseen unbiased testing sets, *i.e.*, the color has not been seen in training set.

| Dataset | CMNIST-75sp | | | CFashion-75sp | | | CKuzushiji-75sp | | |
|---|---|---|---|---|---|---|---|---|---|
| Bias | 0.8 | 0.9 | 0.95 | 0.8 | 0.9 | 0.95 | 0.8 | 0.9 | 0.95 |
| DIR | $10.38_{\pm0.28}$ | $10.14_{\pm0.40}$ | $9.77_{\pm0.18}$ | $16.77_{\pm1.71}$ | $16.51_{\pm3.20}$ | $12.59_{\pm1.61}$ | $10.48_{\pm0.34}$ | $10.33_{\pm0.75}$ | $10.59_{\pm0.95}$ |
| GCN | $36.88_{\pm5.16}$ | $23.07_{\pm4.07}$ | $11.88_{\pm0.33}$ | $59.33_{\pm0.55}$ | $53.65_{\pm0.47}$ | $45.60_{\pm1.06}$ | $36.35_{\pm0.48}$ | $27.88_{\pm0.94}$ | $19.95_{\pm0.67}$ |
| $\text{DisC}_{GCN}$ | $\mathbf{82.73_{\pm1.31}}$ | $\mathbf{77.70_{\pm0.87}}$ | $\mathbf{65.48_{\pm0.76}}$ | $\mathbf{67.9_{\pm1.45}}$ | $\mathbf{68.28_{\pm0.18}}$ | $\mathbf{63.77_{\pm1.37}}$ | $\mathbf{57.80_{\pm2.38}}$ | $\mathbf{51.60_{\pm0.41}}$ | $\mathbf{41.60_{\pm3.94}}$ |
| GIN | $48.93_{\pm2.99}$ | $34.95_{\pm0.86}$ | $14.53_{\pm0.97}$ | $58.88_{\pm0.57}$ | $53.80_{\pm0.52}$ | $48.43_{\pm0.69}$ | $39.25_{\pm0.57}$ | $30.75_{\pm1.45}$ | $22.35_{\pm0.86}$ |
| $\text{DisC}_{GIN}$ | $\mathbf{77.80_{\pm1.33}}$ | $\mathbf{73.00_{\pm0.61}}$ | $\mathbf{58.80_{\pm1.66}}$ | $\mathbf{67.15_{\pm0.79}}$ | $\mathbf{59.98_{\pm0.62}}$ | $\mathbf{51.70_{\pm0.34}}$ | $\mathbf{55.47_{\pm0.98}}$ | $\mathbf{43.20\pm1.36}$ | $\mathbf{31.33_{\pm1.71}}$ |
| GCNII | $53.50_{\pm6.23}$ | $45.52_{\pm2.26}$ | $32.6_{\pm5.66}$ | $58.85_{\pm1.89}$ | $53.98_{\pm0.85}$ | $46.97_{\pm1.38}$ | $39.93_{\pm0.88}$ | $30.33_{\pm1.17}$ | $23.09_{\pm1.83}$ |
| $\text{DisC}_{GCNII}$ | $\mathbf{79.65_{\pm2.13}}$ | $\mathbf{76.63_{\pm1.38}}$ | $\mathbf{60.00_{\pm5.66}}$ | $\mathbf{60.50_{\pm2.77}}$ | $\mathbf{63.05_{\pm2.25}}$ | $\mathbf{61.78_{\pm1.60}}$ | $\mathbf{56.23_{\pm3.45}}$ | $\mathbf{49.10_{\pm2.05}}$ | $\mathbf{41.05_{\pm0.11}}$ |

$\lambda_G$. When $q = 0.1$, it means the GCE loss will nearly reduce to normal CE loss. We can see the performance of $\text{DisC}_{GCN}$ is worse than other scenarios, demonstrating the effectiveness of utilizing GCE loss.

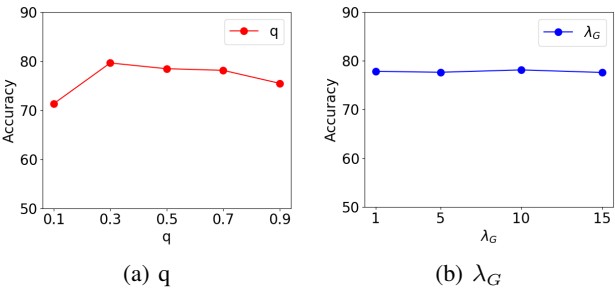

(a) q                                   (b) $\lambda_G$

Figure 5: The hyperparameter experiments of $q$ and $\lambda_G$

## 5.2 Qualitative Evaluation

**Visualization of edge mask.** To better illustrate the significant causal and bias subgraphs extracted by $\text{DisC}_{GCN}$, we visualize the original images, original graph, and corresponding causal subgraph and bias subgraph of CMNIST-75sp with 0.9 bias degree in Fig. 6, where the width of edge represents the value of learned weight $c_{ij}$ or $b_{ij}$. Fig. 6(a) shows the visualization results of testing graphs with the bias (color) that has been seen in the training set. As we can see, our model could discover the causal subgraphs where the most salient edges are in the digital subgraphs. With these causal subgraphs that highlight the structure information of digital, the GNNs will more easily extract this causal information. Fig. 6(b) shows the visualization results of testing graphs with unseen bias. According to the visualization, our model could still discover the causal subgraph outline, indicating our model could recognize causal subgraphs, whether the bias is seen or unseen. The visualization results of CFashion-75sp and CKuzushiji-75sp are shown in App. D.

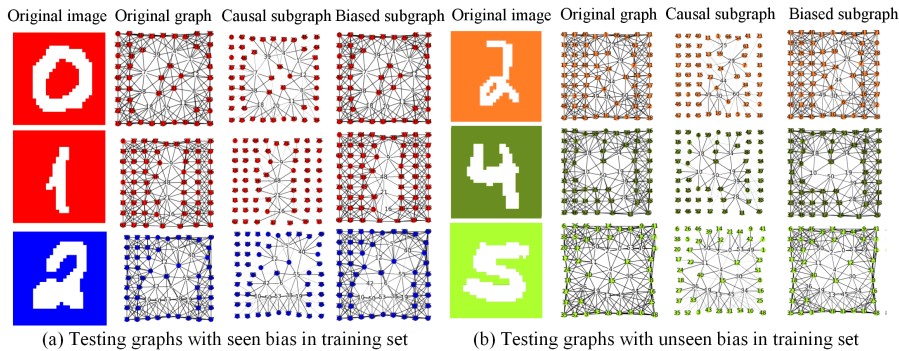

Original image  Original graph  Causal subgraph  Biased subgraph     Original image  Original graph  Causal subgraph  Biased subgraph

(a) Testing graphs with seen bias in training set          (b) Testing graphs with unseen bias in training set

Figure 6: Visualization of subgraphs extracted by DisC. The width of edge is edge weight $c_{ij}$ or $b_{ij}$.

**Projection of disentangled representation.** Fig. 7 shows the projection of latent vectors $z_c$ and $z_b$ extracted from the causal GNN $g_c$ and bias GNN $g_b$ of $\text{DisC}_{GCN}$, respectively, using t-SNE [19] on

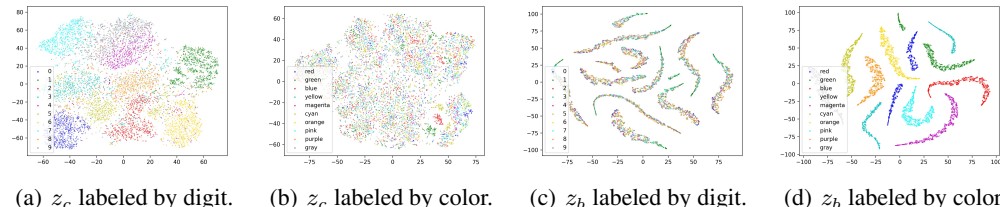

(a) $z_c$ labeled by digit.     (b) $z_c$ labeled by color.     (c) $z_b$ labeled by digit.     (d) $z_b$ labeled by color.

Figure 7: Visualization of $z_c$ and $z_b$ with colors labeled by the digit and bias (color) labels. We observe that $z_c$ and $z_b$ are well clustered according to the groundtruth labels and bias labels, respectively.

CMNIST-75sp. Fig. 7 (a-b) are the projections of $z_c$ labeled by the target labels (digit) and bias labels (color), respectively. Fig. 7 (c-d) are the projections of $z_b$ labeled by the target labels and bias labels, respectively. We observe that $z_c$ are clustered according to the target labels while $z_b$ are clustered with the bias labels. And $z_c$ are mixed with bias labels and $z_b$ are mixed with target labels. The results indicate that DisC successfully learns the disentangled causal and bias representations.

**Transferability of the learned mask.** As our model could extract GNN-independent subgraphs, the learning edge weights can be used to purify original biased graphs. These sparse subgraphs represent significant semantic information and can be universally transferred to any GNNs. To validate this point, we learn the edge mask by $\text{DisC}_{GCN}$ and prune the edges with least $\{0\%, 20\%, 40\%, 60\%\}$ weights while keeping the remaining edge weights. Then we train vanilla GIN and GCNII on these weighted pruned datasets. Fig. 8 is the comparison of the results, where the dashed lines represent the results of base model on original biased graphs and the solid lines represent the performance of GNNs on weighted pruned datasets. The results show that the GNNs trained on the pruned datasets achieve better performances, indicating our learned edge mask has considerable transferability.

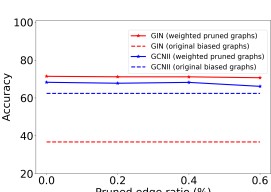

Figure 8: Performance of GIN and GCNII on the weighted pruned graphs found by $\text{DisC}_{GCN}$.

## 6    Conclusion

In this paper, we are first to study the generalization problem of GNNs on severe bias datasets, which is crucial to study the transparently knowledge learning mechanism of GNNs. We analyze the problem in a causal view that the generalization of GNNs will be hindered by entangled representations as well as the correlation between causal and bias variables. To remove the impact from these two aspects, we propose a general disentangling framework, DisC, which extracts causal substructure and bias substructure by two different functional GNNs, respectively. After the representations are well-disentangled, we proliferate the counterfactual unbiased samples by randomly swapping the disentangled vectors. With the new constructed benchmarks, we clearly validate the effectiveness, robustness, interpretability, and transferability of our method.

## Acknowledgments and Disclosure of Funding

This work is supported in part by the National Natural Science Foundation of China (No. U20B2045, 62192784, 62172052, 62002029, 62172052, U1936014). This work is also partially supported by the Natural Sciences and Engineering Research Council (NSERC) Discovery Grant, the Canada CIFAR AI Chair Program, collaboration grants between Microsoft Research and Mila, Samsung Electronics Co., Ltd., Amazon Faculty Research Award, Tencent AI Lab Rhino-Bird Gift Fund and a NRC Collaborative R&D Project (AI4D-CORE-06). This project was also partially funded by IVADO Fundamental Research Project grant PRF-2019- 3583139727. The work of Shaohua Fan is supported by the China Scholarship Council (No.202006470078). The computation resource of this project is supported by Compute Canada[5].

---

[5]https://www.computecanada.ca/

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
