# Supplementary Material of "Debiasing Graph Neural Networks via Learning Disentangled Causal Substructure"

**Shaohua Fan**[1,2][*] **Xiao Wang**[1]**, Yanhu Mo**[1]**, Chuan Shi**[1][†]**, Jian Tang**[2,3,4][†]

[1]Beijing University of Posts and Telecommunications, China

[2] Mila - Québec AI Institute, Canada

[3] HEC Montréal, Canada

[4] CIFAR AI Research Chair

{fanshaohua, xiaowang, moyanhu, shichuan}@bupt.edu.cn, jian.tang@hec.ca

## A    Preliminaries of Causal Inference

### A.1    Structural Causal Models

In order to rigorously formalize our causal assumption behind the dataset, we resort to the Structural Causal Models, or SCM. SCM is a way of describing the relevant features (variables) of a particular problem and how they interact with each other. In particular, an SCM describes how the system assigns values to variables of interest.

Formally, an SCM consists of a set of *exogenous variables* $U$ and a set of *endogenous variables* $V$, and a set of functions $f$ that determines the values of variables in $V$ based on the other variables in the model. Casually, a variable $X$ is a *direct cause* of a variables $Y$ if $X$ exists in the function that determines the value of $Y$. If $X$ is a direct cause of $Y$ or of any cause of $Y$, $X$ is a *cause* of $Y$. Exogenous variables $U$ roughly means that they are external to the model, hence, in most scenarios, we choose not to explain how they are caused. Every endogenous variable is a descendant of at least one exogenous variable. Exogenous variables can only be the root variables. If we know the value of every exogenous variable, with the functions in $f$, we can perfectly determine the value of every endogenous variable. In many cases, we usually assume that all exogenous variables are unobserved variables like noise and are independently distributed with an expected value zero, so we only interest with the interaction with endogenous variables. Every SCM is associated with a *graphical causal model* or simply referred to "casual graph". Causal graph consists of nodes representing the variables in $U$ and $V$, and the direct edges between the nodes representing the functions in $f$. Note the in our SCM in Section 3.2, we only show the endogenous variables we are interested in.

### A.2    $d$-separation/connection

Given an SCM, we are particularly interested in (conditional) dependence information that is embedded in the model. There are three basic relationships of variables in an SCM, *i.e.*, chains, forks and colliders, as shown in Fig. 1. For chains and forks, $X$ and $Y$ would be dependent if $Z$ is not in the conditional set, *i.e.*, the path is unblocked, and vice versa. And for colliders, $X$ and $Y$ would be independent if $Z$ is not in the conditional set, *i.e.*, the path is blocked. Built upon these rules, $d$-separation is a criterion that can be applied in causal graphs of any complexity in order to predict dependencies that are shared by all datasets generated by the graph [2]. Two nodes $X$ and $Y$ are

---

[*]This work was done when the first author was a visiting student at Mila.

[†]Corresponding authors.

36th Conference on Neural Information Processing Systems (NeurIPS 2022).

$d$-separated if every path between them is blocked. If even one path between $X$ and $Y$ is unblocked, $X$ and $Y$ are $d$-connected. Formally, we have following definition of $d$-separation:

**Definition 1** ($d$-**separation [2]**) *A path $p$ is blocked by a set of nodes $Z$ if and only if*

*1. $p$ contains a chain of nodes $A \to B \to C$ or a fork $A \leftarrow B \to C$ such that the middle node $B$ is in $Z$ (i.e., $B$ is conditioned on), or*

*2. $p$ contains a collider $A \to B \leftarrow C$ such that the collision node $B$ is not in $Z$, and no descendant of $B$ is in $Z$.*

With this principle, we could find that the paths **(1) B → G → E → Y** and **(2) B ↔ C → Y** in Section 3.2 are unblocked paths, which would induce unexpected correlation between bias variable $B$ and prediction $Y$.

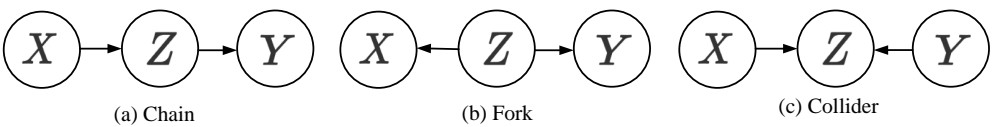

(a) Chain      (b) Fork      (c) Collider

Figure 1: Three basic relations in causal graph.

More knowledge of causal inference, please refer to [2, 5].

## B   Algorithm

---
**Algorithm 1:** Disentangled Casual Substructure Learning

---
**Input**          : graph $G$, label $y$, max iteration $T$, generation iteration $t_{gen}$
**Output**         : Learned edge mask generator MLP, two GNN networks $(g_c, g_b)$, and two classifiers $(C_c, C_b)$
**Initialization** : iteration $t = 0$; Initialize MLP, $(g_c, g_b)$ and $(C_c, C_b)$

1  **while** *not converged* or $t < T$ **do**
2      Extract subgraphs $G_c$ and $G_b$ based on edge mask generator;
3      Encode subgraphs $G_c$ and $G_b$ into $z_c$ and $z_b$ via $g_c(G_c)$ and $g_b(G_b)$;
4      Concatenate $z = [z_c; z_b]$;
5      Update $(g_c, g_b)$ and $(C_c, C_b)$ via $L_D$ in Eq. (6);
6      if $t > t_{gen}$:
7         Randomly swap $z = [z_c; z_b]$ into $z_{unbiased} = [z_c; \hat{z}_b]$;
8         Update $(g_c, g_b)$ and $(C_c, C_b)$ via $L$ in Eq. (8);
9      $t = t + 1$;
10 **end**

---

## C   Experimental Details

### C.1   Datasets details

Table 1: Statistics of Biased Graph Classification Datasets.

| Dataset | Causal subgraph type | Bias subgraph type | #Graphs(train/val/test) | #Classes | #Avg. Nodes | #Avg. Edges | Node feat (dim.) | Bias degree | Difficulties |
|---|---|---|---|---|---|---|---|---|---|
| CMNIST-75sp | Digit subgraph | Color background subgraph | 10K/5K/10K | 10 | 61.09 | 488.78 | Pixel+Coord (5) | 0.8/0.9/0.95 | Easy |
| CFashion-75sp | Fashion product subgraph | Color background subgraph | 10K/5K/10K | 10 | 61.03 | 488.26 | Pixel+Coord (5) | 0.8/0.9/0.95 | Medium |
| CKuzushiji-75sp | Hiragana subgraph | Color background subgraph | 10K/5K/10K | 10 | 52.87 | 423.0 | Pixel+Coord (5) | 0.8/0.9/0.95 | Hard |

We summarize statistics of datasets constructed in this paper in Table 1. Note that the bias degree of validation set is 0.5, we use it to adjust the learning rate during training process. Without loss of any generality, here we subsample original 60K training samples into 10K training samples to make

the training process more efficient. One could easily construct full dataset with our method. Each graph of CFashion-75sp is labeled by the category of fashion product it belongs to and each graph of CKuzushiji-75sp is labeled by one of 10 Hiragana characters. Moreover, we would like to list the map between label and predefined correlated color for all datasets in Table 2. The links for source image datasets are as follows:

1. MNIST: http://yann.lecun.com/exdb/mnist/.
2. Fashion-MNIST: https://github.com/zalandoresearch/fashion-mnist. MIT License.
3. Kuzushiji-MNIST: https://github.com/rois-codh/kmnist. CC BY-SA 4.0 License.

Table 2: Mapping between label and color.

| Label | Color (RGB) | Label | Color (RGB) |
|-------|-------------|-------|-------------|
| 0 | (255, 0, 0) | 5 | (0, 255, 255) |
| 1 | (0, 255, 0) | 6 | (255, 128, 0) |
| 2 | (0, 0, 255) | 7 | (255, 0, 128) |
| 3 | (225, 225, 0) | 8 | (128, 0, 255) |
| 4 | (225, 0, 225) | 9 | (128, 128, 128) |

For unbiased testing dataset with unseen bias used in Table **??**, the RGB value of predefined color set is {(199, 21, 133), (255, 140, 105), (255, 127, 36), (139, 71, 38), (107, 142, 35), (173, 255, 47), (60, 179, 113), (0, 255, 255), (64, 224, 208), (0, 191, 255)}.

### C.2 Experimental setup

For GCN and GIN, we use the same model architectures as [3][3], which have 4 layers, and 146 hidden dimension for GCN and 110 hidden dimension for GIN. And GIN utilizes its GIN0 variant. For GCNII, it has 4 layers and 146 hidden dimension. DIR[4] utilizes the default parameters in original paper for MNIST-75sp dataset. For causal GNN or bias GNN in our model, it has the same architecture with base model. We optimize all models with the Adam [4] optimizer and 0.01 learning rate with for all experiments. The batch-size for all the methods is 256. We train all the models with 200 epochs and set the generation iteration of our method $t_{gen}$ as 100. For our model, we set $q$ of GCE loss as 0.7 and $\lambda_G$ as 10 for all experiments. Our substructure generator is a two-layer MLP, whose activation function is sigmoid function. For StableGNN, we use their GraphSAGE variant. For other baselines, we use their default hyperparameters. LDD[5] has same hyparameters with our model. To better reflect the performance of unbiased sample generation, we take the performances of last step as final results. All the experiments are conducted on the single NVIDIA V100 GPU.

## D   Visualization of CFashion-75sp and CKuzushiji-75sp

Figure 2 and Figure 3 are visualization results of CFashion-75sp and CKuzushiji-75sp dataset. As we can see, our model could also discover reasonable causal subgraphs for these challenging datasets.

## E   Limitations and societal impacts

Our method assumes that the graph consists of causal subgraph and bias subgraph. In reality, it may also exist non-informative subgraphs, which is neither causal nor biased for label. We would like to consider more fine-grained splitting of graphs in the future. When deploying GNNs to real-world applications, especially safety-critical fields, whether the results of GNNs are stable is an important factor. The demands for a stable model are universal and extensive such as in the field of disease prediction [6], traffic states prediction [1], and financial applications [7], where utilizing human-understandable causal knowledge for prediction is necessary.

---

[3]https://github.com/graphdeeplearning/benchmarking-gnns.

[4]https://github.com/Wuyxin/DIR-GNN.

[5]https://github.com/kakaoenterprise/Learning-Debiased-Disentangled

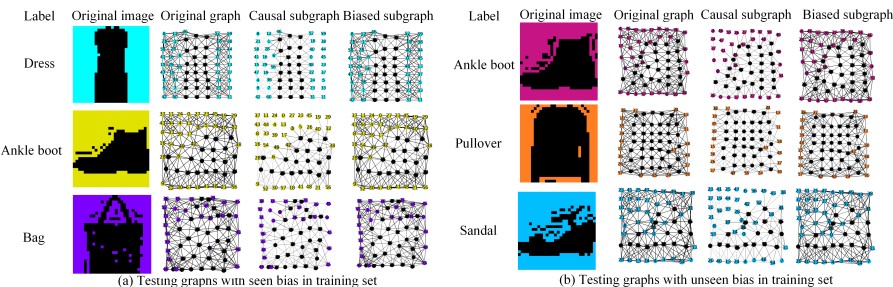

Figure 2: Visualization of subgraphs extracted by the mask generator from CFashion-75sp.

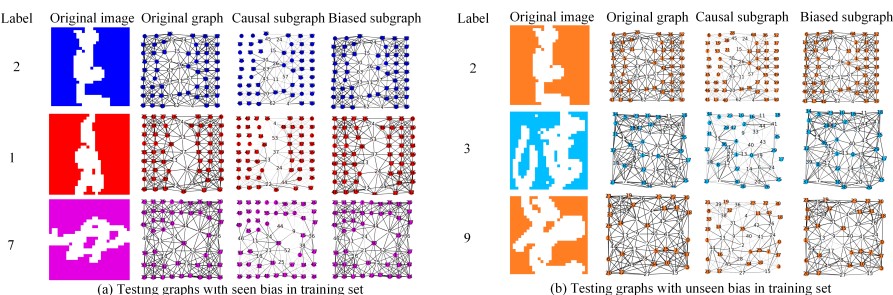

Figure 3: Visualization of subgraphs extracted by the mask generator from CKuzushiji-75sp.