# OpenReview forum: "Debiasing Graph Neural Networks via Learning Disentangled Causal Substructure"
_NeurIPS.cc/2022/Conference — NeurIPS 2022 Accept_

### Official Review · Reviewer_V4kV · 2022-07-10

**Rating:** 8
**Confidence:** 4
**Soundness:** 4 excellent
**Presentation:** 3 good
**Contribution:** 3 good

**Summary:**

This paper pays attention on generalization problem of GNNs on the datasets with severe bias. From their investigation study, an interesting phenomenon shows that the GNNs prefer to utilize spurious correlation to make prediction rather than intrinsic causal information. Based on this observation, the authors analyze the bias problem in causal view and propose the debiasing strategy with the d-connection theory. The proposed framework is reasonable to solve the studied problem, which disentangles the bias and causal substructures by two well-designed functional GNN modules. To benchmark this new problems, three explainable and bias-controllable datasets are constructed. The proposed method significantly outperforms baselines. Meanwhile, the extensive qualitative evaluation could also validate that the proposed method has transparent interpretability.

**Questions:**

1. Recently proposed method, DIR, seems to achieve unsatisfied results on the proposed benchmarks. Could you explain why?
2. The model obtains subgraph by learning the edge mask and do not delete the nodes based on the learned edge weights. I think the model could further remove the bias by deleting the bias nodes.


**Limitations:**

The proposed method may be limited to the problem they study, as they design the model based on the observation from experiments. The model could be further improved by tackling more types of bias, e.g., structure bias.

**Strengths And Weaknesses:**

The paper’s strengths and weaknesses are as follows:
Strengths:
1. The paper proposes an interesting problem, i.e., generalizing GNNs on graphs with severe bias, and also provides an interesting investigation study to show that the spurious correlation will dominate the prediction.
2. The authors utilize SCM to formulate the studied problem. And based on the proposed causal graph, it clearly motivates the methods that make the GNNs get rid of spurious correlation. They first design a parameterized edge mask generator to split the original graph into causal and bias subgraphs. And two GNN modules supervised by bias-aware and causal-aware loss respectively are developed to encode corresponding subgraphs into embeddings. With the learned embeddings, the model further decorrelates the causal and bias embedding by randomly swapping them. Overall, the proposed method is motivated well and technique sound.
3. The experiments are extensive. The paper conducts comprehensive quantitative and qualitative evaluation to validate the effectiveness and various properties of the proposed method. From the results, we can see that the proposed framework could disentangle original bias graph well, resulting in superior performance. Moreover, due to the learned mask, the proposed has appealing interpretability and transferability.
4. The new proposed datasets have various good properties, such as controllable bias degree, easy to explain and visualize. And the proposed benchmarks are desired for the community. The paper also provides the details to reproduce the experiments.
5. Overall, the paper is well-written and easy to follow.

Weaknesses:
Generally, I cannot find major weakness for this paper. This study looks novel in GNN community. I think some parts of model design is inspired by other work from image community, e.g., [20], which may influence its novelty. However, the overall framework is very effective and explainable for the graph data compared with general debiasing method.

---

> ### Author Response · Authors · 2022-08-02
> **Response to Reviewer V4kV**
>
> We sincerely thank the Reviewer for all the comments and it is a great honor for us for your enjoying our paper. We have addressed all your questions below and hope they have clarified all confusion you had about our work.
>
> 1. > I think some parts of model design are inspired by other work from CV community, e.g., [20], which may influence its novelty. However, the overall framework is very effective and explainable for the graph data compared with the general debiasing method.
>
> **Response:** We agree with your comments. The severe bias problem has been studied in the CV community [20], but is largely ignored in the graph community. Our proposed method is very effective on graph data compared with general debiasing method (e.g., LDD[20]) and enjoys the inherent explainability (Fig.5), robustness (Table 2) and Transferability (Fig.7) on graph data.
>
> 2. > Recent proposed method, DIR, seems to achieve unsatisfied results on the proposed benchmarks. Could you explain why?
>
> **Response:** Although DIR is proposed for discovering invariant rationale, they are not designed for severe bias scenarios. Particularly,  generalized CE loss used in our model is very effective to capture the characteristic of bias and reweighted loss is effective in capturing the causal information. Moreover, DIR utilizes a fixed threshold to generate two subgraphs, which is hard to set in real-world dataset. And the same threshold for all graphs is also unreasonable. Hence, the hard threshold of DIR will inevitably achieve suboptimal results compared with directly optimizing the continuous edge weights. We believe these two major reasons would hinder the DIR from discovering the causal rationales, so it cannot achieve satisfied results.
>
> 3. > The model obtains subgraph by learning the edge mask and do not delete the nodes based on the learned edge weights. I think the model could further remove the bias by deleting the bias nodes.
>
> **Response:** Thanks for your constructive suggestion. As deleting the nodes will inevitably require setting the hard threshold,  we decide to use soft edges to split the graph. Because the split subgraphs would highlight the corresponding subgraph by weighted edges, the GNN could learn the corresponding information into their embeddings. And the effectiveness of our method in experiments also supports the rationality of model design.
>
> Thanks again for all your very useful comments. Hope our replies could solve your concerns. If you have more questions, please let us know.

---

### Official Review · Reviewer_UrWX · 2022-07-11

**Rating:** 8
**Confidence:** 5
**Soundness:** 4 excellent
**Presentation:** 3 good
**Contribution:** 4 excellent

**Summary:**

This paper first studies the debiasing problem on GNNs, especially in the severe bias scenarios. They first conduct an investigation study to show than when the bias degree becomes larger, the bias information will dominate the prediction. In these scenarios, existing GNNs will has very poor generalization ability. The paper analyzes the reasons that lead to the failure by a causal graph. This causal graph also inspires two ways to debias GNNs: disentanglement and decorrelation. Based on this high-level idea, the authors propose a GNN disentanglement framework to disentangle the casual and bias substructures. From the extensive experiments on the newly constructed benchmarks, the proposed method outperforms baselines with a large margin. And the proposed model also has very good interpretability, robustness and transferability.

**Questions:**

I am curious about why the GNNs incline to utilize spurious correlation to make prediction rather than causal substructure. Although you have said that bias substructure has simpler structure and is easier to learn, it will be also interesting to conduct more theoretical studies in the GNN learning mechanism view in the further work.

**Ethics Review Area:**

["I don’t know"]

**Limitations:**

The authors point out some limitations (e.g., more fine-grained splitting) and some societal impacts (e.g., the importance of stable model on safety-critical fields). The paper seldom introduces the negative social impacts. And I also could not foresee any negative aspects.

**Strengths And Weaknesses:**

Strengths:
- Originality: The paper first studies the severe bias problem on graphs, which aims to investigate whether the GNNs are vulnerable to the spurious correlation. The paper motivates the proposed method well, both in experimental investigation and causal view. The proposed method could disentangle the desired subgraph well and has clear interpretability owing to the learned edge mask. Moreover, the paper conducts three benchmarks for the direction of study, which have controllable bias degrees and are easier to visualize and explain. All these contributions are original and significant.
-Quality: The studied problem is important, which makes the community being aware of the fundamental learning mechanism of GNN. Can we make more reliable GNNs that utilize human-understandable knowledge to make predictions? The proposed method is solid, which has clear motivation and effective techniques to achieve the goal. The proposed benchmark also has high-quality, which could be standard benchmarks in this direction.
-Clarity: The presentation of this paper is very clear and easy to follow. The contribution of this paper could be easily identified. The paper also provides the code and data to reproduce the experiments.
- Significance: In my opinion, the paper makes significant contributions on problem, technique and benchmark perspectives.

Weaknesses:
1. The presentation can be further improved.
2. It seems that some related works are missing in Section 2, e.g. [1],
[1] Li H, Wang X, Zhang Z, et al. Ood-gnn: Out-of-distribution generalized graph neural network[J]. arXiv preprint arXiv:2112.03806, 2021.

---

> ### Author Response · Authors · 2022-08-02
> **Response to Reviewer UrWX**
>
> We sincerely thank the Reviewer for all the comments and it is a great honor for us for your enjoying our paper. We have addressed all your questions below and hope they have clarified all confusion you had about our work.
>
> 1. > The presentation can be further improved.
>
> **Response:** Thanks for your suggestion. We would like to further polish our paper in the revision.
>
> 2. > It seems that some related works are missing in Section 2, e.g. [1].
>
> **Response:** We would like to include this paper in the revision.
>
> 3. > Conduct more theoretical studies in the GNN learning mechanism view in the further work.
>
> **Response:** We agree with you that conducting more theoretical studies in the GNN learning mechanism view is a very insteresting direction. This study could be conducted on the spectral theory.
>
> Thanks again for all your very useful comments. Hope our replies could solve your concerns. If you have more questions, please let us know.

---

> > ### Comment · Reviewer_UrWX · 2022-08-08
> > **comment after rebuttal**
> >
> > I have read the authors' response. Generally, my concerns are addressed. I think the paper studies a very important problem in GNNs, and the results are also very promising. Therefore, I will keep my rating and vote for acceptance.

---

> > > ### Author Response · Authors · 2022-08-08
> > > **Thank you for your support**
> > >
> > > Thank the reviewer for recognizing the valuable parts of our work. We really appreciate your support.

---

### Official Review · Reviewer_JhdG · 2022-07-13

**Rating:** 4
**Confidence:** 5
**Soundness:** 1 poor
**Presentation:** 2 fair
**Contribution:** 1 poor

**Summary:**

In this work, the authors propose a disentangled GNN framework to learn the causal substructure and bias substructure for generalization. The proposed edge mask generator is to explicitly split the input graph into causal and bias parts. Based on the disentangled representations, the counterfactual unbiased training samples are used to decorrelate causal and bias variables. The authors conduct some experiments on some datasets.

**Questions:**

Besides the concerns above, I still have some questions?
1) How about the results when using GAT as the GNN model besides GCN and GIN?
2) Why randomly permuting the latent vectors from different graphs can guarantee to generate unbiased counterfactual samples?


**Strengths And Weaknesses:**

In this work, the authors propose a disentangled GNN framework to learn the causal substructure and bias substructure for generalization. The proposed edge mask generator is to explicitly split the input graph into causal and bias parts. Based on the disentangled representations, the counterfactual unbiased training samples are used to decorrelate causal and bias variables. The authors conduct some experiments on some datasets.

This work has the following strengths:
1) This work takes the SCM to model the data generation process and focuses on splitting the graph into causal and bias subgraphs. This pipeline can benefit the model interpretability.
2) The model details and framework are present in Figures.
3) The improvements shown in the experiments on some datasets are good.

However, I have the following concerns:
1) The novelty of this paper is so limited, which is the extension of existing works (such as DIR [1]). The technical contributions are straightforward. The problem analysis in section 3.2 is somehow similar to the analysis in [1] (see its appendix C Theory).  Some key part of the designed model has been proposed in DIR [1].  Although they improve upon the CE loss of DIR, the contributions are limited.
2) The expressions are confusing to the readers, and should be carefully revised. The authors propose to learn disentangled causal substructure but it is not explained why the learned representations are disentangled, according to the definition in [2].
3) The proposed method heavily relies on accurately identifying the causal and bias subgraphs at the beginning of the training process, which is hard to be satisfied in practice.
4) The experimental evaluations are not convincing enough to support their claims. For example, some important baselines (such as top-k Pool, SAG Pool, etc.). How to quantitatively verify the learned causal subgraphs have causal relations with the labels? The evaluation datasets are also too weak.
5) The authors claim that the proposed method can not only improve generalization ability but enjoy inherent interpretability, robustness, and transferability. But the experiments are also not enough to support it.
6) Some important literatures such as disentangled graph learning are ignored in the related works and experiments. I sincerely suggest the authors add them into the comparisons.
[1] Discovering Invariant Rationales for Graph Neural Networks.
[2] Towards a Definition of Disentangled Representations.

---

> ### Author Response · Authors · 2022-08-02
> **Response (4/4)**
>
> 6. > Some important literatures such as disentangled graph learning are ignored in the related works and experiments. I sincerely suggest the authors add them into the comparisons.
>
> Thanks very much for your suggestion. As our paper focuses on the debiasing method, we choose to survey and compare with debiasing methods and our base model in our original experimental design, to show the most salient contribution of our model. To our knowledge, the most well-known disentangled method for graph classification FactorGCN [1] is not designed for disentangling causal and bias information. We are happy to compare with FactorGCN  and more related work about disentangled methods have been added in the revision.
> |       Dataset       |   MNIST-75sp   |                |                 |  Fashion-75sp  |                |                | Kuzushiji-75sp |                |                |
> | :-----------------: | :------------: | :------------: | :-------------: | :------------: | :------------: | :------------: | :------------: | :------------: | :------------: |
> |        Bias         |      0.8       |      0.9       |      0.95       |      0.8       |      0.9       |      0.95      |      0.8       |      0.9       |      0.95      |
> |    FactorGCN[1]     | 72.30$\pm$1.18 | 62.35$\pm$5.07 | 42.50$\pm$ 4.91 | 61.23$\pm$1.11 | 53.50$\pm$1.29 | 45.78$\pm$2.40 | 42.87$\pm$1.19 | 32.35$\pm$2.79 | 23.87$\pm$0.12 |
> | $\text{DisC}_{GCN}$ | 82.60$\pm$0.93 | 78.14$\pm$2.14 | 63.47$\pm$5.65  | 66.85$\pm$1.11 | 65.33$\pm$4.70 | 63.93$\pm$1.50 | 55.53$\pm$2.29 | 48.13$\pm$2.59 | 36.63$\pm$1.73 |
>
> From results, our model outperforms SOTA disentangled method for graph classification, showing the effectiveness of our framework to disentangle causal and bias information from graphs.
>
> [1]Yang Y, Feng Z, Song M, et al. Factorizable graph convolutional networks[C]. NeurIPS, 2020.
>
> 7. > How about the results when using GAT as the GNN model besides GCN and GIN?
>
> **Response:** We conduct the experiments of our model utilizing GAT as base model. As GAT is time-consuming,  we only report the results on Colored MNIST-75sp dataset here due to the time limitation.
>
> |       Dataset       | MNIST-75sp      |                 |                 |
> | :-----------------: | --------------- | --------------- | --------------- |
> |        Bias         | 0.8             | 0.9             | 0.95            |
> |         GAT         | 80.03$\pm$ 0.96 | 72.75$\pm$ 0.69 | 53.76$\pm$ 4.02 |
> | $\text{DisC}_{GAT}$ | 84.63$\pm$ 0.38 | 79.43$\pm$ 0.31 | 64.33$\pm$ 6.98 |
>
> According to the results, our model could still make significant improvements based on GAT, indicating the proposed model is general on various GNNs. Considering both effectiveness and efficiency, the base models in our main text are efficient ones.
>
> 8. > Why randomly permuting the latent vectors from different graphs can guarantee to generate unbiased counterfactual samples?
>
> **Response:** We learn causal and bias latent vectors from the observed graph, where the causal and bias vectors $[z_c;z_b]$ will be the latent vectors of subgraphs $G_c$ and $G_b$. $[z_c;z_b]$ inherbit the spurious correlation from the observed graphs. For example, the representations of digit 0 subgraphs are always with red background subgraphs. If we randomly swap the bias representations, the swapped representations will be $[z_c;\hat{z_b}]$, i.e., we randomly assign a background subgraph with digit subgraph. As causal representation $z_c$ and bias representation $\hat{z_b}$ are randomly combined, the statistical correlation between them will be decorrelated. Then the swapped representations $[z_c;\hat{z_b}]$ are the latent vectors of a counterfactual sample that combines causal subgraph $G_c$ and bias subgraph $\hat{G}_b$. In this situation, the causal and bias variables are decorrelated in counterfactual samples, i.e., unbiased, so we call them unbiased counterfactual samples.
>
> Thanks again for all your very useful comments. Hope our replies could solve your concerns. We will add these modifications in the revision. If you have more questions, please let us know. Sincerely hope you could reconsider our work.

---

> ### Author Response · Authors · 2022-08-02
> **Response (3/4)**
>
> 4. > The experimental evaluations are not convincing enough. For example, some important baselines (such as top-k Pool, SAG Pool, etc.) are missing. How to quantitatively verify the learned causal subgraphs have causal relations with the labels? The evaluation datasets are too weak.
>
> **Response:** As our paper focuses on debiasing problem, we choose to compare with debiasing method and our base model in our original experimental design. This experimental design aims to show the most salient contribution of our model. Thanks for your constructive suggestion.  As our dataset preprocessing is based on DGL and the implement of top-k Pool and SAG Pool are not based on DGL, we compare with another popular pooling method DiffPool [1] which has competitive results with them.  The results are shown as follows:
>
> |       Dataset       | MNIST-75sp      |                  |                 | Fashion-75sp    |                 |                 | Kuzushiji-75sp  |                 |                 |
> | :-----------------: | --------------- | ---------------- | --------------- | --------------- | --------------- | --------------- | --------------- | --------------- | --------------- |
> |        Bias         | 0.8             | 0.9              | 0.95            | 0.8             | 0.9             | 0.95            | 0.8             | 0.9             | 0.95            |
> |     DiffPool[1]     | 73.79$\pm$ 0.02 | 66.45$\pm$ 0.78  | 47.12$\pm$ 1.04 | 62.82$\pm$ 0.53 | 57.50$\pm$ 0.39 | 50.86$\pm$ 0.20 | 45.46$\pm$ 0.65 | 36.18$\pm$ 0.19 | 27.45$\pm$ 0.26 |
> | $\text{DisC}_{GCN}$ | 82.60$\pm$ 0.93 | 78.14$\pm$  2.14 | 63.47$\pm$ 5.65 | 66.85$\pm$ 1.11 | 65.33$\pm$ 4.70 | 63.93$\pm$ 1.50 | 55.53$\pm$ 2.29 | 48.13$\pm$ 2.59 | 36.63$\pm$1.73  |
>
> [1] Hierarchical graph representation learning with differentiable pooling. NeurIPS 2018.
>
> As we can see, our model outperforms the graph pooling method, indicating that existing pooling methods could not deal with graph bias.
> All the quantitative results (Table 1 and 2, Fig.4 and 7) could verify the learned causal subgraphs have causal relations with labels. The reason is that the testing set used in evaluation is unbiased, i.e., only the casual subgraphs have the correlation with labels and the bias subgraphs are decorrelated with labels. Hence, the only way to improve the performance is to utilize the causal relation between causal subgraphs with labels to make predictions. Our model achieves better performance, indicating that we could discover such causal relation. Moreover, the visualization of subgraphs extracted by DisC in Fig.5 could also clearly verify our model could utilize the causal subgraph to make predictions.
> All these clear conclusions are due to the well-constructed new datasets. These datasets have controllable bias degrees which enable us to study the severe bias scenario. And they have clear causal relation between causal subgraphs (e.g., digit subgraphs) with labels. Moreover, the bias (i.e., color) and causal subgraphs are easy to visualize. In these aspects, existing benchmarks (e.g., OGB) could not support the study in severe bias scenarios. These reasons are also why we construct these new benchmarks. We believe these datasets are not weak but very suitable for our study. We will release the datasets to benefit further studies in this direction.
>
> 5. > The authors claim that the proposed method can not only improve generalization ability but also enjoy inherent interpretability, robustness, and transferability. But experiments are also not enough to support it.
>
> **Response:** We conduct comprehensive experiments on three datasets with different bias degrees from both quantitative and qualitative perspectives to validate the effectiveness, interpretability, robustness and transferability. Particularly, our paper validates these points from the following experiments:
> - **Generalization ability**: Our method outperforms related baselines with a large margin (Table 1), validating the generalization ability of our method, as the only way to improve the performance is to utilize the causal relations.
> - **Inherent interpretability:** The visualization of edge weights learned by DisC in Fig.5 shows that the causal subgraph part (i.e., digit subgraphs) learned by causal module usually has significantly larger weights than the bias part. The phenomenon indicates that our model could discover causal subgraph successfully, demonstrating the inherent interpretability of the proposed method.
> - **Robustness:** The stable performance of our method on the unseen testing set (Table 2) validates that our model could generalize on unseen bias scenarios, indicating the robustness of our method.
> - **Transferability**: The transferability of the learned mask is validated in Fig.7 that the learned mask of $\text{DisC}_{GCN}$ could be used to prune the edges and transfer to other GNN models.
>
> Therefore, based on the above experiments, we believe our experiments could support our claim.

---

> ### Author Response · Authors · 2022-08-02
> **Response (2/4)**
>
> 2. > The expressions are confusing to the readers, and should be carefully revised. The authors propose to learn disentangled causal substructure but it is not explained why the learned representations are disentangled, according to the definition in [2].
>
> **Response:** Thanks for your kind suggestion. We will carefully modify our paper in the revision.  The definition of disentangled representation in [2] states that:
>
> *"A vector representation is called a disentangled representation with respect to a particular decomposition of a symmetry group into subgroups, if it decomposes into independent subspaces, where each subspace is affected by the action of a single subgroup, and the actions of all other subgroups leave the subspace unaffected."*
>
> The main point of this definition is to decompose subgroups into independent subspaces. Essentially, our model is designed to decompose the causal and bias subgraphs (subgroups) into different representations. Thanks to the elaborately designed loss, the bias module is supervised by the GCE loss to extract the bias subgraph and the causal module is supervised by a reweighted loss
> $W(z) = \frac{CE(C_b(z), y)}{CE(C_c(z), y)+CE(C_b(z), y)}$ to extract the causal subgraph. As the reweight loss puts higher weight on the samples with the larger loss on bias module,  the two GNN modules could pay attention to the different parts of graph and decompose corresponding information into their embeddings. Although we do not strictly constrain the learning subspace to be independent, they are still toward learning decomposed information. Actually, some disentangled methods (e.g., DisenGCN) aim to learn different aspects of latent vectors and do not guarantee that the learned subspaces are independent. We follow their term on disentanglement. And the experiments in Fig.6 could also clearly verify the two kinds of information are disentangled.
>
> 3. > The proposed method heavily relies on accurately identifying the causal and bias subgraphs at the beginning of the training process, which is hard to be satisfied in practice.
>
> **Response:** Several literatures on counterfactual sample generation rely on the well-learned causal mechanism [1][2] . For example, DIR also relies on accurately identifying the causal and bias subgraphs to generate intervened samples. However, compared with DIR, we have made several improvements. First, we utilize the soft edge weights to split the subgraphs rather than a fixed threshold for all graphs used in DIR. Second, due to the hard threshold, the edge weights of DIR could not be directly optimized by the intervened graphs, resulting in suboptimal results. And our edge weights could be directly optimized by the counterfactual samples. Third, the elaborate GCE and reweighted loss could supervise the model to identify two parts more accurately.
>
> [1] Discovering Invariant Rationales for Graph Neural Networks. ICLR 2022.
>
> [2] Counterfactual generative networks. ICLR 2021.

---

> ### Author Response · Authors · 2022-08-02
> **Response to Reviewer JhdG (1/4)**
>
> We sincerely thank the reviewer for spending time and providing valuable feedback. We appreciate all of your suggestions and we have addressed all your questions below by providing our responses as well as our additional experimental results.
>
> 1. > The novelty of this paper is limited.  The problem analysis in section 3.2 is somehow similar to the analysis in [1] (see its appendix C Theory). Some key part of the designed model has been proposed in DIR [1].
>
> **Response:** In general, our causal graph in Fig.2(b) aims to motivate our solution that we should disentangle causal and bias latent vectors and decorrelate them. The solution of DIR is to intervene the graph in input graph space rather than embedding space. Moreover, our model is different from DIR in terms of mask generator, counterfactual sample generation, loss function and overall framework. We would like to clarify the contribution of our paper in terms of causal graph and technical contribution as follows.
>
> - **Causal graph:** Although the data-generation process in Fig.2(a) is similar to causal graph in DIR, our paper emphasizes the severe bias scenario which is hard to be presented in the causal graph. Moreover, our method analyzes the spurious correlation between bias and prediction by the d-connection theory, and DIR does not. The data-generation process for bias graph is similar in several literatures [1-2]. We do not aim to claim that it is our contribution. The debiasing solution presented by Fig. 2(b) is fundamentally different from the theory in DIR (its Appendix C). Our figure aims to express that debiasing method should first disentangle the causal and bias latent variables from observed graphs and decorrelate them in the embedding space, which is exactly consistent with our model design. However, the theory in DIR aims to prove that the DIR objective could discover the invariant causal subgraph based on the intervention on the input graph. For this part, our major difference is that we propose to learn latent casual and bias representations and decorrelate them（generating counterfactual samples）in embedding space, and DIR generates intervened samples in the input space.
> - **Mask generator and counterfactual sample generation:** To generate intervened samples in input space, DIR sets a fixed threshold to split the causal and non-causal subgraph, which is hard to set in real-world data. And the threshold is fixed for all graphs, which is also unreasonable. The quality of the graph generated by DIR will largely rely on the fixed threshold. If DIR could not generate high-quality intervened samples, the overall performance will degenerate. Furthermore, generating real graph data in input graph space is very hard and DIR could only generate the intervened subgraph pair ($\widetilde{c_i}$, $\widetilde{s_i}$), where causal subgraph $\widetilde{c_i}$ and non-causal subgraph $\widetilde{s_i}$ are two disjoint parts and it is unrealistic compared with real graphs. In contrast, our model directly utilizes the edge weights (i.e., $c_{ij}=\sigma (\alpha_{ij}), b_{ij}=1-c_{ij}$) to split the original input graph into two weighted subgraphs and keeps the learned soft edge weights in the filtered subgraphs. And generating counterfactual samples in embedding space based on the learned embeddings of weighted subgraphs will not lead to information loss. These two special designs: without setting a fixed threshold and generating samples in embedding space help the model identify causal subgraphs more accurately.
> - **Loss function**: We have different loss function designs. The generalized CE loss used in our model is very important for capturing the properties of severe bias. However, normal CE loss used in DIR could not capture this property. Furthermore, our reweight strategy on causal module $W(z) = \frac{CE(C_b(z), y)}{CE(C_c(z), y)+CE(C_b(z), y)}$ is totally different from DIR.  Our reweight strategy could make the causal module focus on the information that bias module cannot learn. And the objective of DIR optimizes the model to learn invariant part across multiple interventions on bias subgraphs by an invariant loss. Empirically, our carefully designed loss could achieve much better results.
> - **Overall framework:** DIR utilizes shared GNN for both causal and bias subgraphs. In contrast, our model utilizes separate GNN modules for casual and bias subgraphs, which could better capture the different properties of each kind of subgraphs.
>
> Overall, our proposed model has several key differences from DIR,  leading to the superior performance of our method over DIR.
>
> [1] Discovering Invariant Rationales for Graph Neural Networks. ICLR 2022.
>
> [2] Causal Attention for Interpretable and Generalizable Graph Classification. KDD 2022.

---

> ### Author Response · Authors · 2022-08-09
> **Looking forward to the feedback by Reviewer JhdG**
>
> Dear Reviewer JhdG:
>
> We thank you for taking the time to provide critical comments. We have provided detailed responses that we believe have covered your concerns. As this is the last day for discussion, we kindly remind you that could you check out our reply. We hope to further discuss with you whether or not your concerns have been addressed. Please let us know if you still have any unclear parts of our work.
>
> We are looking forward to your reply.
>
> Best,
>
> Paper6144 Authors.

---

### Official Review · Reviewer_TFtC · 2022-07-13

**Rating:** 3
**Confidence:** 5
**Soundness:** 2 fair
**Presentation:** 2 fair
**Contribution:** 1 poor

**Summary:**

This paper studies the generalization problem of GNNs. The authors analyze the problem that the generalization of GNNs will be hindered by entangled representations, as well as the correlation between causal and bias variables. An edge masker is proposed to extract causal and bias substructures, which are further utilized to construct counterfactual unbiased samples and help GNN to capture causal relations. Experiments aim to validate the method in terms of debiasing, transferability, and interpretability.

**Questions:**

Please kindly see the weaknesses above.

**Limitations:**

N/A


After Rebuttal Update: My rating will not change.

**Strengths And Weaknesses:**

Strengths
1. The research problem of the generalization of GNNs is interesting to the community.
2. The preliminary study and analysis (Section 3) further clarify the motivations of this paper.
3. The experiments show the effectiveness of the method on three datasets, including quantitative and qualitative evaluations.

Weaknesses
1. Line 73 " To our knowledge, we first study the generalization problem of GNNs in a more challenging yet practical scenario, i.e., the graphs are with severe bias." is overclaimed to me. Some works are overlooked[1-4].
2. It seems that the paper differs from DIR[2] only in some minor details: it changes the cross-entropy to generalized cross-entropy, etc. Why does DIR perform so poorly in Table 1? Why is DIR missing in Table 2?
3. The experiment evaluations miss some necessary parts. The used datasets are all super-pixel datasets converted from images, but graph benchmarks on generalization are totally ignored in the experiments (such as OGB, etc.). And the hyperparameter analysis is also missing, which is also important since some hyperparameters could influence the performance significantly.
4. The related works about disentangled graph learning are missing.
5.  The main idea seems similar to DIR[2]. The authors are expected to clarify and highlight the difference between DIR and this paper.
[1]Zhu, Qi, et al. "Shift-robust gnns: Overcoming the limitations of localized graph training data."
[2]Wu, Ying-Xin, et al. "Discovering invariant rationales for graph neural networks."
[3]Fan, Shaohua, et al. "Debiased graph neural networks with agnostic label selection bias."
[4]Zhang, Shengyu, et al. "Stable Prediction on Graphs with Agnostic Distribution Shift."

---

> ### Author Response · Authors · 2022-08-02
> **Response (4/4)**
>
> 4. > The related works about disentangled graph learning are missing.
>
> **Response: **Our original thought is that we study debiasing problem on GNNs, so we mainly focus on the debiasing methods. Although there are several disentangled graph learning methods have been proposed[1-3], they are not designed for disentangling bias and causal information and only [3] can be used for graph classification. Thanks for your suggestion, and we will add these methods into our related work. And we also add the experiments for method [3] in the rebuttal to validate whether exisiting disentangled method could deal with debiasing problem.
>
> |       Dataset       |   MNIST-75sp   |                |                 |  Fashion-75sp  |                |                | Kuzushiji-75sp |                |                |
> | :-----------------: | :------------: | :------------: | :-------------: | :------------: | :------------: | :------------: | :------------: | :------------: | :------------: |
> |        Bias         |      0.8       |      0.9       |      0.95       |      0.8       |      0.9       |      0.95      |      0.8       |      0.9       |      0.95      |
> |    FactorGCN[3]     | 72.30$\pm$1.18 | 62.35$\pm$5.07 | 42.50$\pm$4.91  | 61.23$\pm$1.11 | 53.50$\pm$1.29 | 45.78$\pm$2.40 | 42.87$\pm$1.19 | 32.35$\pm$2.79 | 23.87$\pm$0.12 |
> | $\text{DisC}_{GCN}$ | 82.60$\pm$0.93 | 78.14$\pm$2.14 | 63.47$\pm$ 5.65 | 66.85$\pm$1.11 | 65.33$\pm$4.70 | 63.93$\pm$1.50 | 55.53$\pm$2.29 | 48.13$\pm$2.59 | 36.63$\pm$1.73 |
>
> From the results, our model outperforms SOTA disentangled method for graph classification, showing the effectiveness of our framework to disentangle causal and bias information from graphs.
>
> [1] Disentangled graph convolutional network. ICML, 2019
> [2] Independence promoted graph disentangled networks. AAAI, 2020.
> [3] Factorizable graph convolutional networks. NeurIPS, 2020.
>
> 5. > The main idea seems similar to DIR[2]. The authors are expected to clarify and highlight the difference between DIR and this paper.
>
> **Response:** We would like to highlight the differences between our paper with DIR from the following perspectives:
>
> - **At the problem level:** We study the severe bias scenario and DIR could not handle this scenario well. As we investigated in Sec.3, the spurious correlation will be very strong and dominate the prediction. DIR is not designed to capture this property of bias. Based on the observation on the property of severe bias, we propose our model to deal with such a challenging debiasing problem.
> - **At the method level:** Our method has several key substantial differences from DIR in terms of mask generator, counterfactual sample generation, loss function and overall framework (please refer to Q2 for more details). All these differences contribute to our superior performance over DIR.
> - **At the dataset level**: To meet the requirement of study on the severe bias, we construct three new datasets. These datasets have controllable bias degrees and clear causal relation between causal subgraphs with labels, which is very suitable for the study of debiasing problem on graphs. These datasets have the important properties that existing datasets do not have. They will be the important supplement of existing datasets on biased graphs. We will release the datasets to benefit further studies in this direction.
>
> Thanks again for all your constructive comments. Hope our responses could solve your concerns. We have modified the corresponding part in the revision. If you have more questions, please let us know. Sincerely hope you could reconsider our work.

---

> ### Author Response · Authors · 2022-08-02
> **Response (3/4)**
>
> |       Dataset       |   MNIST-75sp   |                |                |  Fashion-75sp  |                |                | Kuzushiji-75sp |                |                |
> | :-----------------: | :------------: | :------------: | :------------: | :------------: | :------------: | :------------: | :------------: | :------------: | :------------: |
> |        Bias         |      0.8       |      0.9       |      0.95      |      0.8       |      0.9       |      0.95      |      0.8       |      0.9       |      0.95      |
> |         DIR         | 10.38$\pm$0.28 | 10.14$\pm$0.40 | 9.77$\pm$0.18  | 16.77$\pm$1.71 | 16.51$\pm$3.20 | 12.59$\pm$1.61 | 10.48$\pm$0.34 | 10.33$\pm$0.75 | 10.59$\pm$0.95 |
> | $\text{DisC}_{GCN}$ | 82.73$\pm$1.31 | 77.70$\pm$0.87 | 65.48$\pm$0.76 | 67.90$\pm$1.45 | 68.28$\pm$0.18 | 63.77$\pm$1.37 | 57.80$\pm$2.38 | 51.60$\pm$0.41 | 41.60$\pm$3.94 |
>
> From the results, we can see that DIR still could not achieve satisfied results in the unseen bias scenario.
>
> 3. > The used datasets are all super-pixel datasets converted from images, but graph benchmarks on generalization are totally ignored in the experiments (such as OGB, etc.). And the hyperparameter analysis is also missing, which is also important since some hyperparameters could influence the performance significantly.
>
> **Response:** To study the severe bias scenario, it requires the dataset has controllable bias degrees and easy to be explained causal relations. These properties will ensure the correctness and explainability of evaluation that we could debias and leverage causal relations to make predictions. As far as we know, existing benchmarks (e.g., OGB) do not meet this requirement. Based on this reason, we construct three datasets with controllable bias degrees and clear causal subgraphs. These datasets have the important properties that OGB does not have. They will be the important supplement of existing datasets on biased graph. We will release these datasets to promote further studies in this direction.
>
> Moreover, our model does not have many parameters to finetune. We set the degree of amplifying bias $q$ in GCE loss as 0.7 and the importance of generation component  $\lambda_G$ as 10 for all experiments, and find that it already achieves satisfied results. The phenomenon further validates the robustness of our model. According to your suggestion, we are happy to provide the hyperparameter experiments of $q$ and $\lambda_G$ on Colored MNIST-75sp dataset with 0.9 bias degree. For $q$, we fix $\lambda_G=10$ and vary $q$ from $\{0.1, 0.3, 0.5, 0.7, 0.9\}$. For $\lambda_G$, we fix $q=0.7$ and vary $\lambda_G$ from $\{1, 5, 10, 15\}$.
>
> |         $q$         | 0.1        | 0.3            | 0.5             | 0.7             | 0.9            |
> | :-----------------: | ---------- | -------------- | --------------- | --------------- | -------------- |
> | $\text{DisC}_{GCN}$ | 71.33+4.34 | 79.64$\pm$1.35 | 78.47$\pm$ 1.85 | 78.14$\pm$ 2.14 | 75.49$\pm$1.88 |
>
> |     $\lambda_G$     | 1              | 5               | 10              | 15             |
> | :-----------------: | -------------- | --------------- | --------------- | -------------- |
> | $\text{DisC}_{GCN}$ | 77.85$\pm$2.14 | 77.66$\pm$ 1.90 | 78.14$\pm$ 2.14 | 77.62$\pm$1.25 |
>
> From the results, overall our model achieves stable performance across different values of $q$ and $\lambda_G$. We notice that when $q=0.1$ (it means the GCE loss will nearly reduce to normal CE loss) the performance of $\text{DisC}_{GCN}$ is worse than other scenarios (i.e., increase the degree of amplifying bias $q$), demonstrating the effectiveness of utilizing GCE loss.

---

> ### Author Response · Authors · 2022-08-02
> **Response (2/4)**
>
> 2. > It seems that the paper differs from DIR[2] only in some minor details: it changes the cross-entropy to generalized cross-entropy, etc. Why does DIR perform so poorly in Table 1? Why is DIR missing in Table 2?
>
> **Response**: First, we study a different problem from DIR. Our study focuses on the severe bias scenario, but DIR fails to capture such bias properties, leading to unsatisfied performance. To solve this problem, our model has several key substantial differences from DIR, which could be concluded from the following perspectives:
> 1. **Mask generator**: Our mask generator utilizes the soft edge weights (i.e., $c_{ij}=\sigma (\alpha_{ij})$, $b_{ij}=1-c_{ij}$) to split the original input graph into two weighted subgraphs and keeps the learned edge weights in the filtered subgraphs. However, DIR sets a fixed threshold for edge weights to split two subgraphs that if the edge weights higher than the threshold the edge will be causal subgraph and lower will be bias subgraph. Then DIR discards the learned weights to get two unweighted subgraphs. First, the fixed threshold is hard to set in real-world data, and setting the same threshold for all graphs is also unreasonable. Second,  keeping the edge learned weights in our model will benefit in learning more informative embeddings. Third, in the optimization aspect, the hard threshold of DIR will inevitably lead to suboptimal results compared with directly optimizing the continuous edge weights.
>   2. **Counterfactual unbiased sample generation**: We generate counterfactual unbiased samples in embedding space by swapping the latent bias vectors, but DIR generates intervented samples in the input space by replacing the non-causal part with the candidate non-causal subgraphs in a memory bank. The quality of the intervened graph generated by DIR will heavily rely on the fixed threshold. If DIR could not generate high-quality intervened samples, the overall performance will degenerate.  Furthermore, generating real graph data in input space is very hard and DIR could only generate the intervened subgraph pair $(\widetilde{c}_i, \widetilde{s}_i)$, where causal subgraph $\widetilde{c}_i$ and non-causal subgraph $\widetilde{s}_i$ are two disjoint parts and it is unrealistic compared with real graphs. Generating counterfactual samples in embedding space based on the learned embeddings of weighted subgraphs will not lead to such information loss. These two special designs: without setting a fixed threshold and generating samples in embedding space in our model  will help our method generate high-quality samples.  Moreover, as we do not need to maintain a memory bank, our method will be more memory efficient and easy to implement in embedding space.
>   3. **Loss function**: We have different loss designs. The generalized CE loss used in our model is very important to capture the characteristic of severe bias. However, normal CE loss used in DIR could not capture this property. Moreover, our reweight strategy on causal module $W(z) = \frac{CE(C_b(z), y)}{CE(C_c(z), y)+CE(C_b(z), y)}$ is totally different from DIR. Our reweight strategy could make the causal module focus on the information that bias module cannot learn. And the objective of DIR optimizes the model to learn invariant part across multiple interventions on bias subgraphs by an invariant loss. Empirically, our carefully designed loss could achieve much better results.
>   4. **Overall framework:** DIR utilizes shared GNN for both causal and bias subgraphs. In contrast, our model utilizes separate GNN modules for casual and bias subgraphs respectively, which could better capture the different properties of each kind of subgraphs.
>
> Overall, our proposed method has significant differences from DIR. All these differences will contribute to the significant improvements of our method over DIR in Table 1. We believe the most important two factors resulting in the unsatisfied results of DIR are its loss function and fixed threshold for all graphs, which could not help the model to capture the property of bias and identify causal subgraph precisely. Moreover, we guarantee the correctness of performing DIR. We have carefully finetuned the key hyperparameters (e.g., edge weight threshold, learning rate, embedding dim and batch size) and the code for reproducing DIR are in the code and data link in the main text. As DIR does not achieve competitive results in the main experiments of Table1 and Table 2 is an anylsis experiment on the robustness of our method in unseen bias scenario, we select the most competitive baselines in Table 2. We are happy to provide the DIR experiments in Table 2 during rebuttal as follows:

---

> ### Author Response · Authors · 2022-08-02
> **Response to Reviewer TFtC (1/4)**
>
> We sincerely thank the reviewer for constructive reviews. We would like to address the concerns of the reviewer by providing our responses as well as our additional experimental results.
>
> 1. > Line 73 " To our knowledge, we first study the generalization problem of GNNs in a more challenging yet practical scenario, i.e., the graphs are with severe bias." is overclaimed. Some works are overlooked[1-4].
>
> **Response**: First, literatures [1, 3, 4]  focus on the node classification task, but our paper studies the graph classification task. Different from work [2] which studies a general bias problem on graph classification, our paper emphasizes the scenario that graphs have **severe** bias. The severe bias scenario is more challenging and has different properties compared with lighter bias scenarios. As we have investigated in Sec.3, although causal substructure could determine labels perfectly, in severe bias scenarios, the GNNs lean to utilize the easier-to-learn bias information to make predictions rather than the inherent causal signals, and bias substructure will finally **dominate** the prediction. This unique bias property in the severe bias scenario is what we want to highlight, however, DIR is not designed to capture this property. Our method is proposed to fully capture the properties of severe bias by the GCE loss on bias module. Because we could capture bias properties well, we can utilize the proposed reweight loss to extract the complementary information (i.e., causal subgraph) accurately.  Moreover, from the experimental results, our method outperforms the method dealing with general bias on graph classification (i.e., DIR) with a large margin.

---

> ### Author Response · Authors · 2022-08-09
> **Looking forward to the feedback by Reviewer TFtC**
>
> Dear Reviewer TFtC:
>
> We thank you for taking the time to provide critical comments. We have provided detailed responses that we believe have covered your concerns. As this is the last day for discussion, we kindly remind you that could you check out our reply. We hope to further discuss with you whether or not your concerns have been addressed. Please let us know if you still have any unclear parts of our work.
>
> We are looking forward to your reply.
>
> Best,
>
> Paper6144 Authors.

---

### Author Response · Authors · 2022-08-02
**Revision modification**

We thank all the reviewers for their constructive feedback and suggestions. Based on this feedback we have added a substantial amount of new results and analysis. We believe that the quality of our paper has been improved. The main additions are summarized below:

- The major concerns of Reviewer TFtC and JhdG, differences between our paper with DIR, are adequately discussed in the replies.

- Add two baselines. One is a classical graph pooling method DiffPool and another is SOTA disentangled GNN method for graph classification FactorGCN. (See Table 1 in revision and corresponding replies)

- Add the results of DIR in Table 2. (Also in the replies for reviewer TFtC)

- Add the results of hyperparameter experiments. (See Appendix D and the replies for reviewer TFtC)

- Add the results of our method's variant on GAT. (See the replies for reviewer jhdG)

- Add a paragraph of related work on disentangled GNN methods in Section 2.

We reply to each reviewer's concerns in detail below their reviews. Please kindly check out them. Looking forward to your further replies.

---

### Meta-Review · Area_Chair_8ZcE · 2022-08-28

**Recommendation:** Accept
**Confidence:** Less certain

**Metareview:**

The paper proposes a GNN framework where the causal substructure and the bias substructure are disentangled by a mask generator and separate GNNs.  SOTA results on artificially generated severely biased data are reported.

Reviewers raised concerns mainly on the novelty, e.g., from [2, 20], insufficient experiments, missing baselines, and experiments only on artificially generated data based on images (not really graph data).  The authors addressed those concerns mostly well.

Although two reviewers kept their rejecting scores, they didn't further raise criticisms after rebuttal, and I find no good reason for rejection from their reviews.

My concern that remains after the rebuttal is about the data.  The authors argue that existing benchmark graph datasets don't have severe bias, which is why they generated artificially biased data by using image data, which should not necessarily be treated as graphs.  This makes me wonder if the proposed method is really useful in practice.  Namely, are there some application scenarios where severe bias is expected on graph data?   If so, readers can expect that the authors would prepare real-world graph data that show severe bias (without manipulation), on which the proposed method outperforms the SOTA methods.  Can readers expect this in the author's near future follow-up work?  I strongly recommend the authors to discuss this point in the final version.


**Award:**

No

---

### Decision · Program_Chairs · 2022-09-14

Accept